# Grain boundary engineering for efficient and durable electrocatalysis

Xin Geng [1,6] ✉, Miquel Vega-Paredes [1,6], Zhenyu Wang [1] ✉, Colin Ophus [2], Pengfei Lu[3], Yan Ma [1,4], Siyuan Zhang [1], Christina Scheu [1], Christian H. Liebscher [1] & Baptiste Gault [1,5] ✉

Grain boundaries in noble metal catalysts have been identified as critical sites for enhancing catalytic activity in electrochemical reactions such as the oxygen reduction reaction. However, conventional methods to modify grain boundary density often alter particle size, shape, and morphology, obscuring the specific role of grain boundaries in catalytic performance. This study addresses these challenges by employing gold nanoparticle assemblies to control grain boundary density through the manipulation of nanoparticle collision frequency during synthesis. We demonstrate a direct correlation between increased grain boundary density and enhanced two-electron oxygen reduction reaction activity, achieving a significant improvement in both specific and mass activity. Additionally, the gold nanoparticle assemblies with high grain boundary density exhibit remarkable electrochemical stability, attributed to boron segregation at the grain boundaries, which prevents structural degradation. This work provides a promising strategy for optimizing the activity, selectivity, and stability of noble metal catalysts through precise grain boundary engineering.

In heterogeneous catalysis, the catalysts' activity and stability are critical factors for practical applications[1]. Integration of strain, through incorporating microstructural defects or doping, is an effective strategy for synthesizing highly active nanocatalysts[2–11]. Lattice strain modulates the surface electronic structures, i.e., d-band center and width, which can be used for optimizing the binding energy of reaction intermediates and hence reducing the overall energy barrier for specific reactions[12,13]. In addition, microstructural defects offer distinctive coordination environments and surface configurations, which also tune the electronic structures and reaction pathways[3,5–7]. Grain boundaries (GBs) are two-dimensional structural defects that can effectively synergize these strategies: they can induce local lattice strain[3,7,12], atomic step ridges[6], and offer unique active sites for reactions like the four-electron oxygen reduction reaction (ORR)[14–18]. For

instance, Maillard et al. demonstrated that Pt–Ni nanocrystals with increased surface distortion exhibit superior four-electron ORR activities compared to their less defective counterparts[16–18]. However, due to the challenge of introducing GBs in nanocatalysts in a controlled manner, the understanding of how GBs modulate catalytic activity and their potential to catalyze other reactions remains limited. Existing strategies for tuning GB density modify also the size, shape and morphology of nanocatalysts, hampering the interpretation of the relationship between GBs and catalytic activity[14]. Our understanding is also limited by incomplete characterization of GB type distribution, their atomic arrangement and local composition[14]. This lack of understanding leads to a negligence of the potential of GBs to regulate not only the catalytic activity but also selectivity towards certain pathways. For instance, having an active catalyst for oxygen reduction[14,15] that is

[1]Max Planck Institute for Sustainable Materials, Düsseldorf, Germany. [2]National Center for Electron Microscopy, Molecular Foundry, Lawrence Berkeley National Laboratory, Berkeley, CA, USA. [3]School of Energy and Power Engineering, Huazhong University of Science and Technology, Wuhan, China. [4]Department of Materials Science and Engineering, Delft University of Technology, Delft, the Netherlands. [5]Department of Materials, Royal School of Mines, Imperial College London, London, UK. [6]These authors contributed equally: Xin Geng, Miquel Vega-Paredes. ✉e-mail: x.geng@mpie.de; z.wang@mpie.de; b.gault@mpie.de

also selective towards its two-electron pathway would enable the electrochemical production of hydrogen peroxide ($H_2O_2$), a chemical widely used in various industries[19–21].

In addition, the stability of GBs during electrocatalysis is also debated. Some studies suggest that Pt GBs are stable during four-electron ORR[14,22], while others report poor stability of GBs in polycrystalline PtCo nanowires compared to single-crystal counterparts[15]. Atoms at GBs are thermodynamically unstable due to their high energy, which can lead to structural degradations and catalyst deactivation[15]. Inconsistent stability reports may be due to local compositional variations and trace impurities at GBs[23], which are challenging to detect with conventional techniques but have been reported to effectively stabilize these defects. For instance, boron (B) is reported to stabilize the GBs of bulk polycrystals by lowering their energy[24,25], enhancing overall structural integrity. However, its effect on GB stabilization in nanocatalysts remains unexplored. Advanced atomic-scale characterization methods, such as atom probe tomography (APT)[23], high-angle annular dark-field (HAADF) scanning transmission electron microscopy (STEM)[26], and 4D-STEM[27], are essential for investigating the local composition and structure near GBs, thus advancing our understanding of their catalytic roles.

Here, we have achieved the formation of GB-rich nanoparticles assemblies through the collision, attachment, and coalescence of gold (Au) NPs' surfaces devoid of capping agents (Fig. 1a). The NP collisions are driven by the continuous bubbling of $H_2$ gas into the solution, and the GB density can be controlled by the gas flow rate that modifies the collision frequency. The introduction of GBs leads to pronounced lattice distortions, resulting in local strain near the GB regions, stacking faults, dislocations and atomic step ridges. Furthermore, an expansion of the crystal lattice parameter and a simultaneous reduction in coordination number are observed with the formation of an increased number of GBs. The Au nanoassemblies exhibit high efficiency in catalyzing the two-electron ORR, a more than two orders of magnitude increase in mass and specific activity with respect to GB-poor Au NPs showcasing the pivotal role of GBs in promoting high reactivity and selectivity. Moreover, the stability of Au nanoassemblies has been found to depend on the concentration of boron species that segregate to the GBs to lower their free energy, inhibiting diffusion and annihilation of GBs. Heteroatom engineering at GBs effectively stabilizes active sites and preserves high two-electron ORR activity with minimal degradation over more than 100 hours.

## Results

### Grain boundary rich nanoparticle assemblies

The introduction of $H_2$ gas at a high flow rate (300 standard cubic centimeters per minute, sccm) into a solution containing citrate-capped Au NPs (Supplementary Figs. S3–S4) initiates the attachment of Au NPs into porous, nanoscale assemblies (denoted as H-Au NAs, Supplementary Fig. S5), in which the NPs act as building blocks that maintain their initial size (Supplementary Fig. S6). The assembly of Au NPs comprises three distinct stages. Firstly, under mildly alkaline conditions (pH=10), the initially capped citrate ligands undergo gradual detachment from the NP surface (Supplementary Fig. S7) due to their instability (Supplementary Fig. S8)[28], and as a consequence, $OH^-$ ions can adsorb onto the NP surface. During this stage, the steric repulsion between Au NPs originating from the citrate ligands is alleviated, although electrostatic repulsion persists due to the adsorption of $OH^-$ ions (Supplementary Fig. S9a–c)[29]. Secondly, the continuous bubbling of an excess amount of hydrogen gas into the Au NP solution facilitates competitive adsorption between $OH^-$ ions and H atoms. This results in the gradual replacement of surface-bound $OH^-$ ions by H atoms, thereby transforming the negatively charged NP surface into a nearly neutral state and eliminating electrostatic repulsion between Au NPs (Supplementary Fig. S9c). Thirdly, the introduction of $H_2$ gas into the NP solution induces turbulence and convection currents, thereby

increasing the frequency of NP collisions. Upon approaching each other, two NPs can undergo spatial rotation, followed by attachment and coalescence at regions devoid of ligands to form a GB. The primary driving force for this spontaneous attachment is the elimination of bare regions lacking ligand protection, possessing high surface energies, thereby reducing the total energy of the system[30]. Other structural defects such as step sites, stacking faults, and dislocations may arise due to imperfect orientation during the attachment process. Furthermore, NP attachment typically results in anisotropic growth, giving rise to nanowire morphology[31]. If other gases (e.g., Ar, $N_2$) are used instead of $H_2$, the NP attachment does not occur (Supplementary Fig. S9d–i) due to the electrostatic repulsions between $OH^-$ capped NPs' surfaces. Each H-Au NAs consists of individual Au crystals, which can be regarded as their NP building blocks, connected by GBs (Fig. 1b). Among a randomly selected set of 22 GBs (Fig. 1b and Supplementary Fig. S10), we have identified 16 Σ3 GBs, 1 Σ11 GB, 2 Σ27 GBs, 1 Σ33 GB based on coincident site lattice theory (CSL)[32], and 2 low-angle GBs (LAGBs). We also observed five-fold twin boundaries in Au NAs (Fig. 1b), likely formed through high-energy GB decomposition and partial dislocation slipping during NP attachment[33]. These twin boundaries enhance structural stability by minimizing surface and interfacial energies, resulting in a stable and energetically favorable five-fold symmetric structure.

To obtain a statistical analysis of the GBs present in H-Au NAs, we have employed four-dimensional scanning-transmission electron microscopy (4D-STEM) (Fig. 1c–e, Supplementary Figs. S11–14)[34]. Among ~600 GBs, H-Au NAs contain ~7% LAGBs and ~93% high-angle GBs (HAGBs), including 31% Σ3 GBs along with other CSL GBs (Fig. 1e). Although a trace number of Au NPs possess low-energy Σ3 GBs (Supplementary Fig. S15), the vast majority of them are found to be free of GBs (Supplementary Fig. S16), implying that most of the GB defects are formed during the assembly process. Therefore, the GB density of Au NPs is set to $0 \, \mu m^{-1}$ in the follow up discussion. During the collision event between two NPs initially randomly oriented, they can reorient to lower energy orientations[33], explaining why lower energy GBs (i.e., Σ3) are more frequently observed. As the assemblies grow there are lower degrees of freedom left, and higher energy HAGBs form as well. Beyond GBs, H-Au NAs contain stacking faults and dislocations (Fig. 1b), which can be remnants of the rotation process of multiple NPs merging[33].

Remarkably, the shapes of individual NP building blocks within H-Au NAs have changed significantly when compared to the isolated Au NPs (Figs. 1b, 2a). Spherical NPs have formed nanoassemblies with nearly polygonal grains, implying that collisions among NPs or their posterior reorientation have induced important deformations in the structure of these building blocks. The surface of the H-Au NAs exhibits a high number of atomic step ridges (Fig. 2a–c, Supplementary Fig. S17), which are likely a result of surface reconstruction taking place during the assembly process. Relative to terrace sites, step sites possess lower coordination numbers, thus exhibiting higher catalytic activity[7,35–37]. Moreover, the GB terminations at the surface form line defects with a distinct coordination environment (Fig. 2d), which can also enhance the catalytic activity[7]. Numerous triple junctions (i.e., a line defect where three GBs converge) are also observed on H-Au NAs (Fig. 2a, Supplementary Fig. S18). These line defects propagate to the surface of the NAs, creating a point defect that can act as a catalytically active site. Additionally, deviations of the relaxed atomic positions are observed in the nanoassemblies, as seen by the distortions on the GBs structural units in the vicinity of the triple junctions (Fig. 2d, e) and by the presence of lattice strain. High levels of strain are found next to the higher energy GBs (i.e., Σ9, Σ27) and triple junctions (Fig. 2f–h). This results in strain-rich surface regions next to the defects surface terminations, which can optimize the binding energy of reaction intermediates. Strain maps from the 4D-STEM data in a larger area (hundreds of square nanometers) reveal a similar uneven distribution

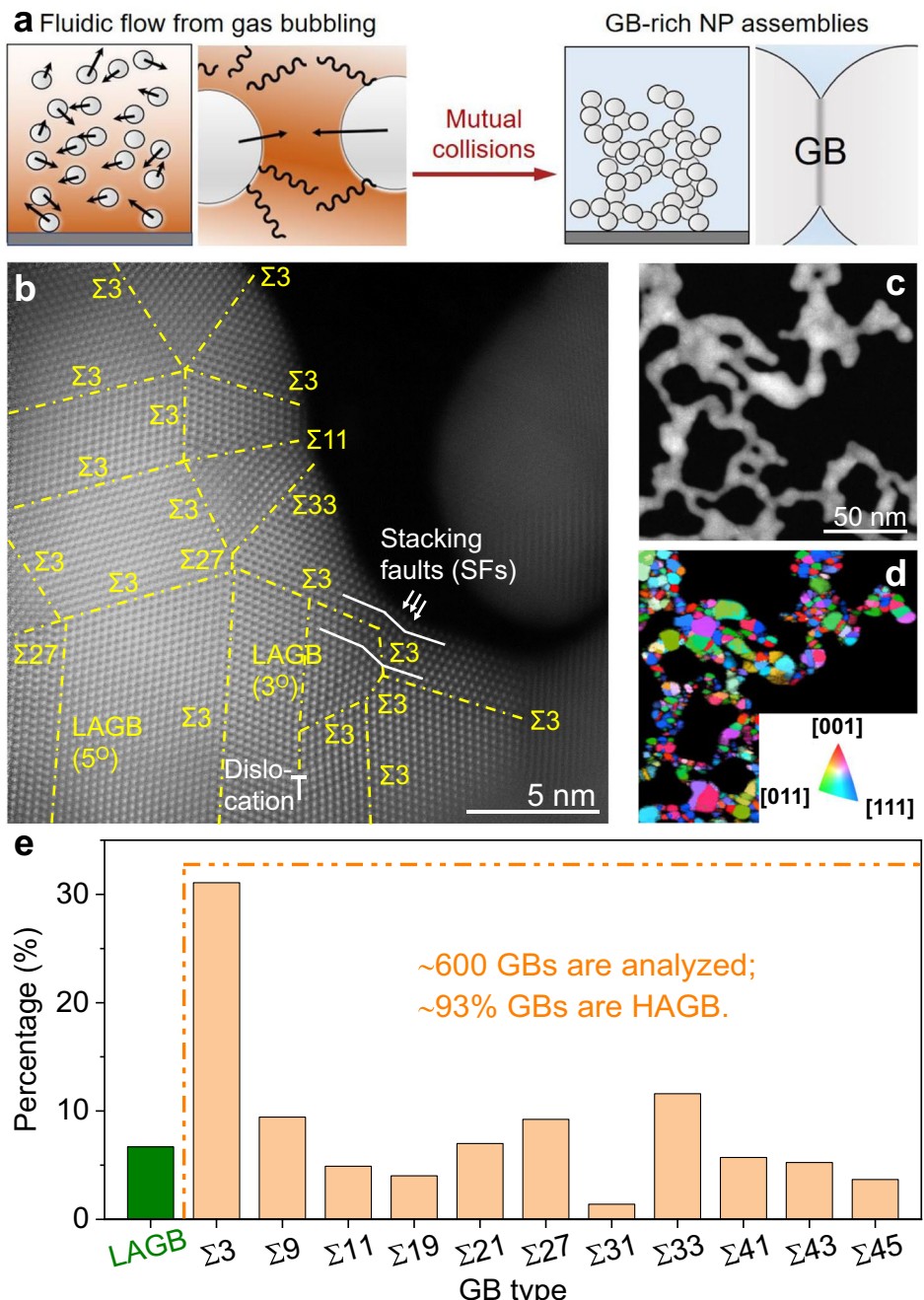

**Fig. 1 | Synthesis of GB-rich nanoassemblies. a** Illustration depicting the formation of GB-rich Au NAs through the collision, attachment, and coalescence of NPs' surfaces devoid of capping agents facilitated by $H_2$ gas bubbling. **b** High resolution HAADF-STEM micrograph of the GBs between the Au NP building blocks within H-Au NAs. **c** HAADF-STEM image and (**d**) grain orientation maps from the corresponding 4D-STEM dataset for H-Au NAs. The inset shows the color code for the out-of-plane coordinates. **e** Histogram plots of the GB types derived from ~600 GBs in the 4D-STEM data. LAGB refers to low-angle GBs with a misorientation of less than 15 degrees, while HAGB denotes high-angle GBs with a misorientation exceeding 15 degrees. Source data are provided as a Source Data file.

of the strain, with higher strain levels located in the regions between the NP building blocks (Fig. 2i–k, Supplementary Figs. S12–S14).

## Tuning of grain boundary density

We now demonstrate how the $H_2$ gas bubbling can modulate the GB density in the NAs, by using both a low flow-rate of 30 sccm (L-Au NAs) and a medium flow-rate of 100 sccm (M-Au NAs). With increasing flow-rate, the NAs transitioned in morphology from a loosely connected network to a network with more ramifications (Fig. 3a–e, Supplementary Figs. S19–S20). This can be attributed to an increase in the frequency of collisions among NPs induced by the increasing fluid flow[38,39], since it leads to a homogenization of growth probabilities across various sites, thereby contributing to the augmented number of ramifications. The GB surface density is defined as the ratio of the total GB surface length to the exposed surface area. For calculating it, we relied on the surface area reduction during the assembly process assuming that each NP attachment leads to the creation of a GB (see methods)[40] and that crystallite size does not change significantly during the assembly process, as confirmed by the size distribution histograms calculated from the STEM images (Supplementary Fig. S21). The surface area is quantified by monitoring the reduction of gold oxides using cyclic voltammetry under equivalent catalyst load

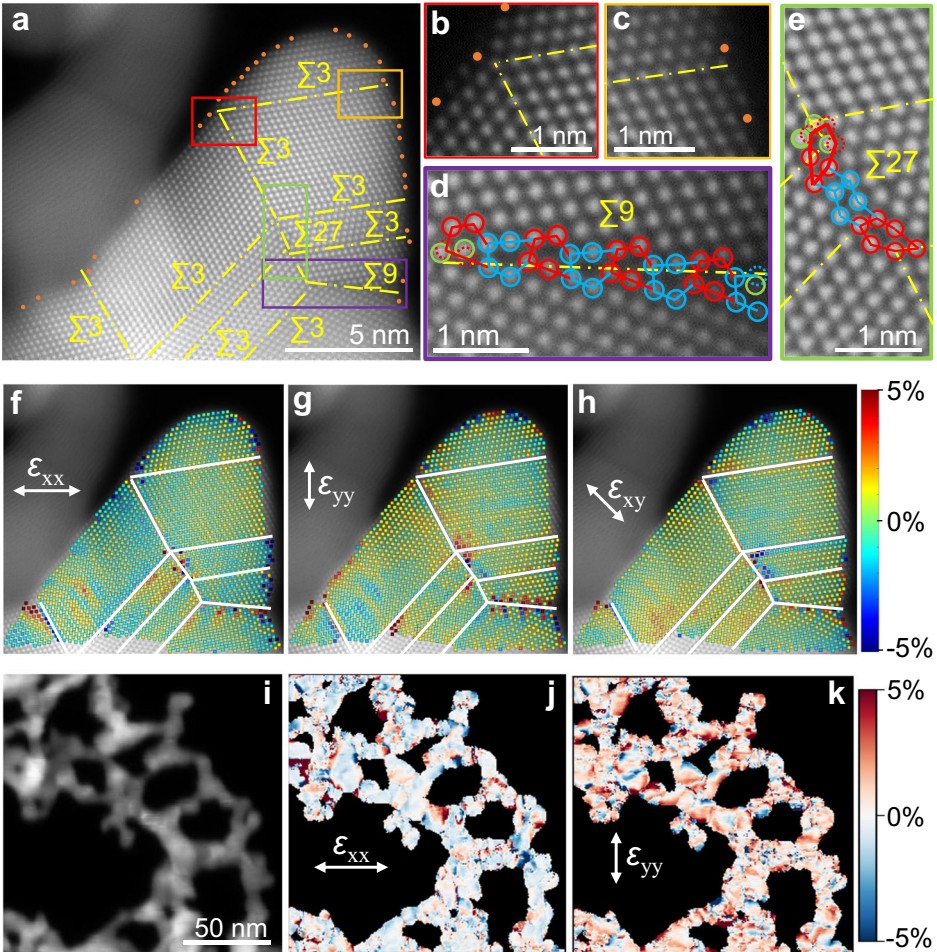

**Fig. 2 | Defects and strain characterization for H-Au NAs. a** HAADF-STEM image of GBs and undercoordinated surface atoms (orange dots) in H-Au NAs. Enlarged regions showing the (**b**, **c**) GB surface terminations and structure of (**d**) a $\sum9/(221)$ and (**e**) $\sum27/(552)$ GB. **f**–**h** Real space strain maps along the x ($\varepsilon_{xx}$), y ($\varepsilon_{yy}$) and xy ($\varepsilon_{xy}$) directions. Notice how on the higher energy GBs elevated values of strain are found. **i** Virtual annular dark field image from the 4D-STEM data and (**j**–**k**) corresponding relative strain maps ($\varepsilon_{xx}$ and $\varepsilon_{yy}$), showing a similar inhomogeneous strain distribution.

(Fig. 3f). With this method, we estimate that the GB surface density within the Au NAs progressively increases with the gas flow rate from 98 $\mu m^{-1}$ at 30 sccm (L-Au), 160 $\mu m^{-1}$ at 100 sccm (M-Au), to 235 $\mu m^{-1}$ at 300 sccm (H-Au) (Fig. 3g). This trend aligns with the morphology evolution evaluated by STEM (Fig. 3a–e), which indicates a higher degree of NP interconnectivity at elevated flow rates, and therefore a higher GB density (Fig. 3g). Brunauer-Emmett-Teller (BET) measurements indicate that L-Au NAs, M-Au NAs, and H-Au NAs possess large specific surface areas and high porosity (Supplementary Fig. S22, Table S1). Among these, L-Au NAs exhibits the largest specific surface area, pore diameter, and pore volume, while H-Au NAs has the smallest. The specific surface area and porosity of M-Au NAs are intermediate between those of L-Au NAs and H-Au NAs. The trends in BET-measured specific surface areas of Au NAs are consistent with electrochemical measurements. 4D-STEM measurements indicate that both L-Au NAs and M-Au NAs have a similar GB type distribution to H-Au NAs, comprising approximately 10% LAGBs and 90% HAGBs, including about 30% $\sum3$ GBs alongside other CSL GBs (Supplementary Fig. S23). The rate of hydrogen gas bubbles did not affect the GB type distribution. The same facets are found in the Au NPs before the assembly process and in the NAs (Supplementary Figs. S10, S16).

The X-ray diffraction (XRD) peak positions of (111) and (200) crystallographic planes of Au NAs exhibit a monotonous shift towards lower angles (Fig. 3h) indicating a gradual expansion of the lattice with increasing gas flow rate. This is further confirmed by analyzing the

Fourier-transformed Extended X-ray Absorption Fine Structure (EXAFS) spectra in the frequency domain, that reveal an increase in the bond length of the NAs with an increasing flow rate (Fig. 3i, Table S2). The lattice expansion derived from EXAFS analysis of L-Au NAs, M-Au NAs, and H-Au NAs are approximately 2%, 4%, and 6%, respectively. Interestingly, we found a linear relationship between GB density and the degree of lattice expansion. Additionally, the EXAFS spectra indicate that there is a sequential reduction in coordination number (Fig. 3i–j, Table S2), transitioning in the order of Au NPs (~12), L-Au NAs (~10.9), M-Au NAs (~9.9), and H-Au NAs (~8.7). The decline in coordination numbers can be ascribed to the increased density of defects[5,41], including atomic step ridges, dislocations and stacking faults, likely stemming from an elevated collision rate. This reduction in coordination number may result in an enhancement of the reactivity of atoms within Au NAs[42].

## Two-electron oxygen reduction reaction performance

Having demonstrated that we can tune the GB surface density, lattice parameter and coordination number of Au NAs via the $H_2$ flow rate control, we test their catalytic activity towards two-electron ORR to study the influence of GBs in their catalytic performance. Developing an efficient, selective and durable catalyst in acidic media for this reaction has represented a challenge[43], yet it is critical to secure an environmentally-friendly way to synthesize $H_2O_2$[20,21], a product with extensive industrial applications[21]. The electrocatalytic performance

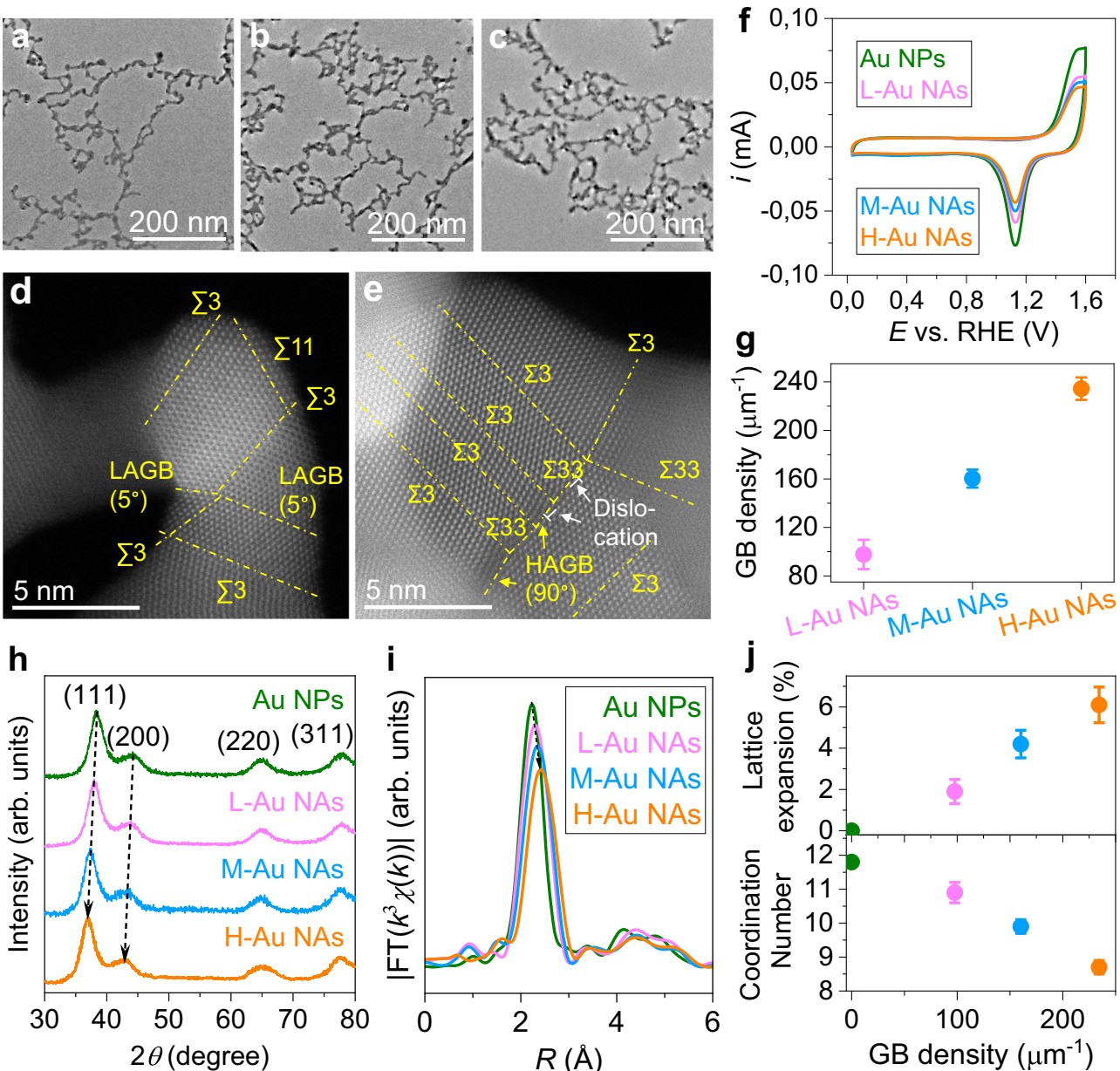

**Fig. 3 | GB density tuning by varying H₂ flow rate.** TEM images of Au NAs prepared from different gas flow rates, encompassing (**a**) a low-flow-rate of 30 sccm, denoted as L-Au NAs, (**b**) a medium-flow-rate of 100 sccm, designated as M-Au NAs and (**c**) a high-flow-rate of 300 sccm, identified as H-Au NAs. HAADF-STEM image of the GBs in (**d**) L-Au NAs and (**e**) M-Au NAs. LAGB indicates low-angle GBs with a misorientation of less than 15 degrees, while HAGB refers to high-angle GBs with a misorientation greater than 15 degrees. **f** Cyclic voltammograms in 0.1 M HClO₄ at scan rate of 100 mV s⁻¹. **g** Relationship between gas flow rates and GB surface density (see Supporting Information for more details). GB density measurements were conducted at least three times, with average values reported. Error bars represent the standard deviation. **h** XRD patterns of L-Au NAs, M-Au NAs and H-Au NAs, where a noticeable shift toward lower angles is observed for the (111) and (200) peaks, as highlighted by the black arrows indicating an expansion of the lattice. **i** Fourier-transformed EXAFS spectra for Au L3-edge for Au NPs, L-Au NAs, M-Au NAs, and H-Au NAs. **j** Correlation between GB surface density and lattice expansion/coordination number calculated from the Fourier-transformed EXAFS spectra. Error bars indicate the standard deviation from measurements taken at least three times. Source data are provided as a Source Data file.

towards the two-electron ORR is assessed for both Au NPs and Au NAs employing a rotating ring-disk electrode (RRDE) with a calibrated collection efficiency of 36% in an oxygen-saturated 0.1 M HClO₄ electrolyte (Fig. 4a, Supplementary Fig. S24). In comparison to Au NPs, Au NAs exhibit a significant enhancement in the onset potential (i.e., lower overpotential) required to achieve a current density of 0.1 mA cm⁻² (Fig. 4a), accompanied by a notable increase in H₂O₂ selectivity (Fig. 4b). As the applied potential varies, the selectivity for two-electron ORR in Au NPs diminishes, while Au NAs maintain consistent selectivity (Fig. 4b). This reduction in selectivity for Au NPs is due to the aggregation of under-coordinated surface atoms, which are more

prone to aggregation than more-coordinated atoms[44,45], leading to fewer under-coordinated atoms with higher two-electron ORR selectivity remaining exposed (Table S2–S3). In contrast, the coordination number of Au NAs remains unchanged after the two-electron ORR test (Table S2–S3), likely because the network geometry of Au NAs and their multiple contact points with the carbon support reduce motion and aggregation tendencies. The lower overpotential and H₂O₂ selectivity of H-Au NAs make them superior two-electron ORR catalysts, outperforming all existing H₂O₂ catalysts in acidic media (Fig. 4c, Table S4). Moreover, we found that there exists a linear relationship between the onset potential/selectivity of two-electron ORR and GB

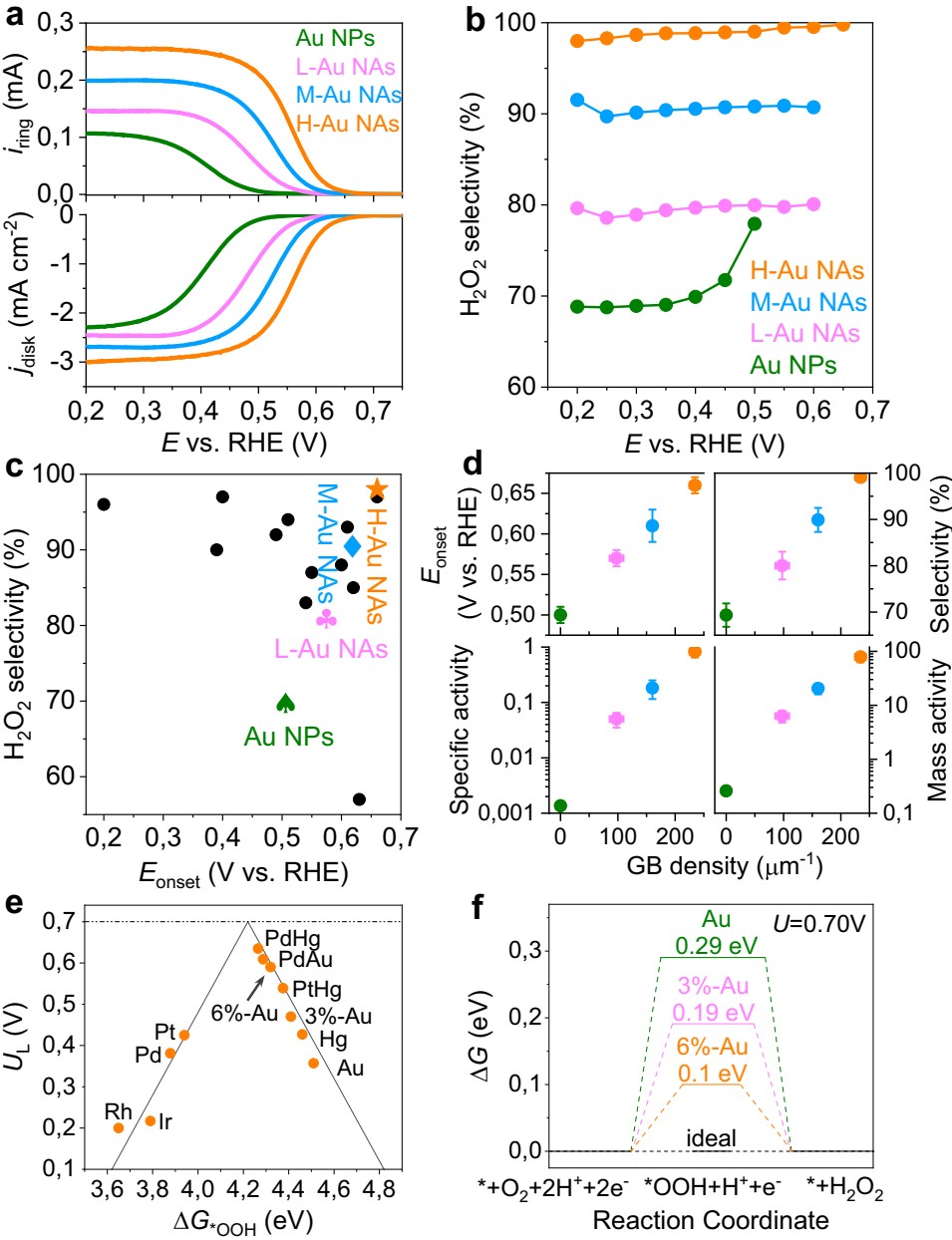

**Fig. 4 | Two electron ORR activity measurements and rationalization. a** Linear sweep voltammetry of Au NPs, L-Au NAs, M-Au NAs and H-Au NAs recorded at 1600 rpm and a scan rate of 5 mV s$^{-1}$ in 0.1 M HClO$_4$, together with the detected H$_2$O$_2$ currents on the ring electrode (upper panel) at a fixed potential of 1.2 V vs. RHE. **b** Calculated H$_2$O$_2$ selectivity during potential sweep. **c** Comparison of the onset potential and H$_2$O$_2$ selectivity of Au catalysts developed in this study and the state-of-the-art H$_2$O$_2$ catalysts in 0.1 M HClO$_4$ from the literature (Table S4). **d** Correlation between GB density and the two-electron ORR activity, including onset potential, H$_2$O$_2$ selectivity, specific activity, and mass activity. Error bars indicate the standard deviation based on at least three independent measurements. **e** The calculated ORR activity volcano relationship between the limiting potential ($U_L$) and the free energy of *OOH ($\Delta G_{*OOH}$) for the two-electron pathway to H$_2$O$_2$. **f** Calculated reaction coordinate diagrams for the Au and lattice-expanded Au. Source data are provided as a Source Data file.

surface density (Fig. 4d). When contrasted with Au NPs, H-Au NAs exhibit an enhancement exceeding two orders of magnitude in both specific and mass activity with respect to the two-electron ORR (Fig. 4d). It suggests that the exceptional performance for H-Au NAs may be attributed to the highest GB surface density, largest lattice expansion, and lowest coordination number as a consequence of NP collisions.

To rationalize the two-electron ORR high activity of the Au NAs, we calculated the volcano plot using the limiting potential as performance indicator (Fig. 4e). This potential is defined as the maximum potential at which both the one-electron reduction of O$_2$ to *OOH and the subsequent one-electron reduction of *OOH to H$_2$O$_2$ is

energetically favorable. Metals with a strong binding energy to the *OOH intermediate specie such as Pd or Pt have an excellent ORR activity but tend to directly reduce O$_2$ to H$_2$O via the four-electron pathway, resulting in a poor selectivity towards the two-electron ORR[46]. On the contrary, metals like Au or Hg that interact weakly with *OOH can selectively produce H$_2$O$_2$, but tend to have a low catalytic activity[47]. The optimization of catalytic performance in the two-electron ORR of H-Au NAs relies critically on the interplay between lattice expansion, coordination number, and GBs. (1) Expanding the lattice of Au improves the binding energy between Au and *OOH (Fig. 4e). Strain-free Au sites positioned on the right side of the volcano plot exhibit a low affinity for *OOH adsorption, impairing O$_2$

adsorption. Lattice expansion significantly enhances *OOH adsorption strength, shifting it closer to the volcano plot's peak. Notably, Au with a 6% lattice expansion—similar to that observed in H-Au NAs—shows a free energy deviation of only 0.1 eV from the ideal state, markedly lower than the 0.29 eV deviation seen in unstrained Au (Fig. 4f). Additionally, regions near GBs experience high strain levels (Fig. 2), which can further boosts *OOH adsorption energy and align it more closely with the volcano plot's maximum. (2) The coordination number also significantly influences two-electron ORR performance (Supplementary Fig. S25). When it has a value of 9 or larger, the *OOH adsorption energy is too weak (located on the right side of the volcano plot; Supplementary Fig. S25b), reflecting a diminished affinity for *OOH and hindering $O_2$ adsorption. Reducing the coordination number enhances *OOH adsorption strength, progressively shifting it towards the volcano plot's peak (Supplementary Fig. S25b). A coordination number of 8 achieves a free energy deviation of just −0.01 eV from the ideal state (Supplementary Fig. S25c), positioning *OOH adsorption energy nearly at the plot's peak. If the coordination number is further decreased, the affinity towards *OOH is too high, which can hinder product desorption and cause catalyst poisoning, thereby slowing subsequent reactions. Thus, a coordination number of approximately 8 is optimal for achieving superior two-electron ORR activity, highlighting its role in optimizing both activity and selectivity. 3) GBs also play a crucial role in two-electron ORR activity (Supplementary Fig. S26). On GB-free Au surfaces, the *OOH adsorption energy is too low (positioned on the right side of the volcano plot; Supplementary Fig. S26b), signifying low *OOH affinity and restricted $O_2$ adsorption. In contrast, GB atoms have a significantly higher *OOH adsorption energy, which falls on the left side of the volcano plot, reflecting a higher affinity compared to GB-free sites (Supplementary Fig. S26b). The Σ3 GB model, in particular, shows a minimal free energy deviation of −0.01 eV from the ideal state (Supplementary Fig. S26c), placing *OOH adsorption energy at the volcano plot's peak. Conversely, *OOH adsorption energies for Σ9 GB and Σ27 GB models exceed the optimal range, which may facilitate too much $O_2$ dissociation and adversely affect two-electron ORR selectivity. Analysis of surface valence band photoemission spectra reveal a progressive increase in the d-band center energy, following the order: Au NPs (−4.57 eV) < L-Au NAs (−4.46 eV) < L-Au NAs (−4.33 eV) < L-Au NAs (−4.16 eV) (Supplementary Fig. S27). Both the increased lattice expansion and the reduced coordination number jointly contribute to an upward shift in the d-band center[42], which correlates with an elevation in the adsorption energy of *OOH on Au surfaces. Experimental measurements of the d-band center (Supplementary Fig. S27), in conjunction with theoretical computations, consistently prove that lattice expansion serves to increase the adsorption strength of *OOH on the surface of Au. This increment, together with the high density of GBs and other structural defects, results in H-Au NAs being a highly effective catalyst for the two-electron ORR.

To assess the durability, Au NPs and NAs are subjected to a continuous 100-h operation in the RRDE setup (Fig. 5a). Notably, Au NPs exhibit a rapid decay in both disk and ring currents, with ~55% activity loss within the initial 40-h of operation at an applied potential of 0.35 V vs. RHE. In contrast, Au NAs demonstrate significantly improved durability, with a considerably smaller decline in activity relative to Au NPs, and their $H_2O_2$ selectivity remains largely unaffected (Fig. 5b). The diminished durability of Au NPs relative to Au NAs can be attributed to their susceptibility to motion, aggregation, and Ostwald ripening process during electrochemical reactions (Supplementary Fig. S28)[44,45]. The network geometry of Au NAs, coupled with multiple contact points with the carbon support, likely reduces such motion and aggregation tendencies[48]. Additionally, it may hinder Ostwald ripening typically observed in spherical NPs, thereby contributing to their excellent durability. The overall morphology and the dimensions of Au NAs supported on carbon exhibit minimal alteration during the

durability tests (Supplementary Fig. S29). However, NAs with a similar morphology experience a different degradation behavior. In particular, Au NAs synthesized at higher flow rates (H-Au NAs) show an enhanced durability compared to those synthesized at lower flow rates (M-Au NAs and L-Au NAs). This can be explained by the preservation of structural defects such as GBs, dislocations and atomic step ridges. Notably, L-Au NAs manifest a substantial decrease in both GBs and other defects (Fig. 5c) while most of these defects, including GBs, dislocations and steps are conserved in H-Au NAs (Fig. 5d, Supplementary Fig. S30). This could be attributed to differences in the chemical composition at the GBs of these Au NAs. Indeed, segregation of heteroatoms has been shown to decrease the free energy of GBs in NAs[10,23], and more generally in metallic and ceramic materials[10,23,49,50]. X-ray photoelectron spectroscopy (XPS) analysis reveals a shift toward lower binding energy in the Au 4$f$ spectra of Au NAs with increasing flow rates (Supplementary Fig. S31), indicating a fine-tuning of their electronic structure[2]. Furthermore, trace amounts of boron stemming from boric acid used in the synthesis have been found within Au NAs by XPS (Supplementary Fig. S31). However, the precision of XPS in quantifying B and its spatial resolution are limited, posing challenges to achieving precise quantitative analyses.

To overcome this limitation and accurately ascertain the spatial distribution of B, APT is employed (Supplementary Fig. S32). The near-atomic-scale APT analysis of Au NPs reveals that B is located on their surfaces (-10 at.%.) and no B is detected inside the particles (Supplementary Fig. S33). In the atom probe tomograms for the different NAs (Fig. 5e–j, Supplementary Figs. S34–S37), individual NP building blocks can be readily discerned, and the concave regions were used to identify GB joining two particles. No B is detected on the surface of the NAs. B atoms appear all distributed in the vicinity of GBs (Fig. 5f, i, Supplementary Fig. S36). 1D composition profiles along the arrow indicate an atomic concentration of approximately 2%, 5%, and 8% for B at GBs within L-Au NAs, M-Au NAs, and H-Au NAs, respectively (Fig. 5g, j). The flow rate-dependent B concentration within GBs in Au NAs suggests a correlation between the collision rate among NPs and the detachment rate of surface-bound B species prior to NP attachment, strongly suggesting that B species are kinetically trapped within the GBs. Despite the high B concentration in the GBs, the overall B concentrations in L-Au NAs, M-Au NAs, and H-Au NAs are smaller, of 0.14 at.%, 0.36 at.%, and 0.57 at.%, respectively. To determine whether B segregation contributes to the high two-electron ORR activity and durability, B was intentionally excluded during the synthesis to produce B-free Au NAs under a high flow rate of $H_2$ gas of 300 sccm (denoted as B-free H-Au NAs). B-free Au NAs exhibit a highly porous network where NP building blocks are interconnected predominantly by Σ3 GBs (Supplementary Fig. S38), similar in morphology (Supplementary Fig. S38a, Fig. 1c), lattice expansion (Table S5), and coordination number (Table S5) to H-Au NAs. The two-electron ORR activity of B-free Au NAs (Supplementary Fig. S39a, b) is comparable to that of H-Au NAs (Fig. 5a, b), indicating that B segregation does not account for the high two-electron ORR activity observed in H-Au NAs (Supplementary Fig. S40). Durability tests for two-electron ORR revealed a current decay of 43.1% for B-free H-Au NAs after 100 hours (Supplementary Fig. S39c,d), whereas H-Au NAs showed a significantly lower decay of only 3.8% (Fig. 5a, b). Post-durability tests characterization indicates substantial structural changes in B-free H-Au NAs, marked by an increased coordination number and decreased lattice expansion (Table S5). In contrast, H-Au NAs structure was maintained, highlighting the essential role of B in enhancing structural stability and consequent ORR durability. To further explore the impact of B on GB stability, we conducted DFT simulations using a B-segregated Au Σ3 GB model at varying B concentrations (Supplementary Fig. S41a). The results demonstrated that increasing B concentration at GBs leads to more negative GB strengthening energy (Supplementary Fig. S41b), suggesting that higher B concentrations enhance GB strength,

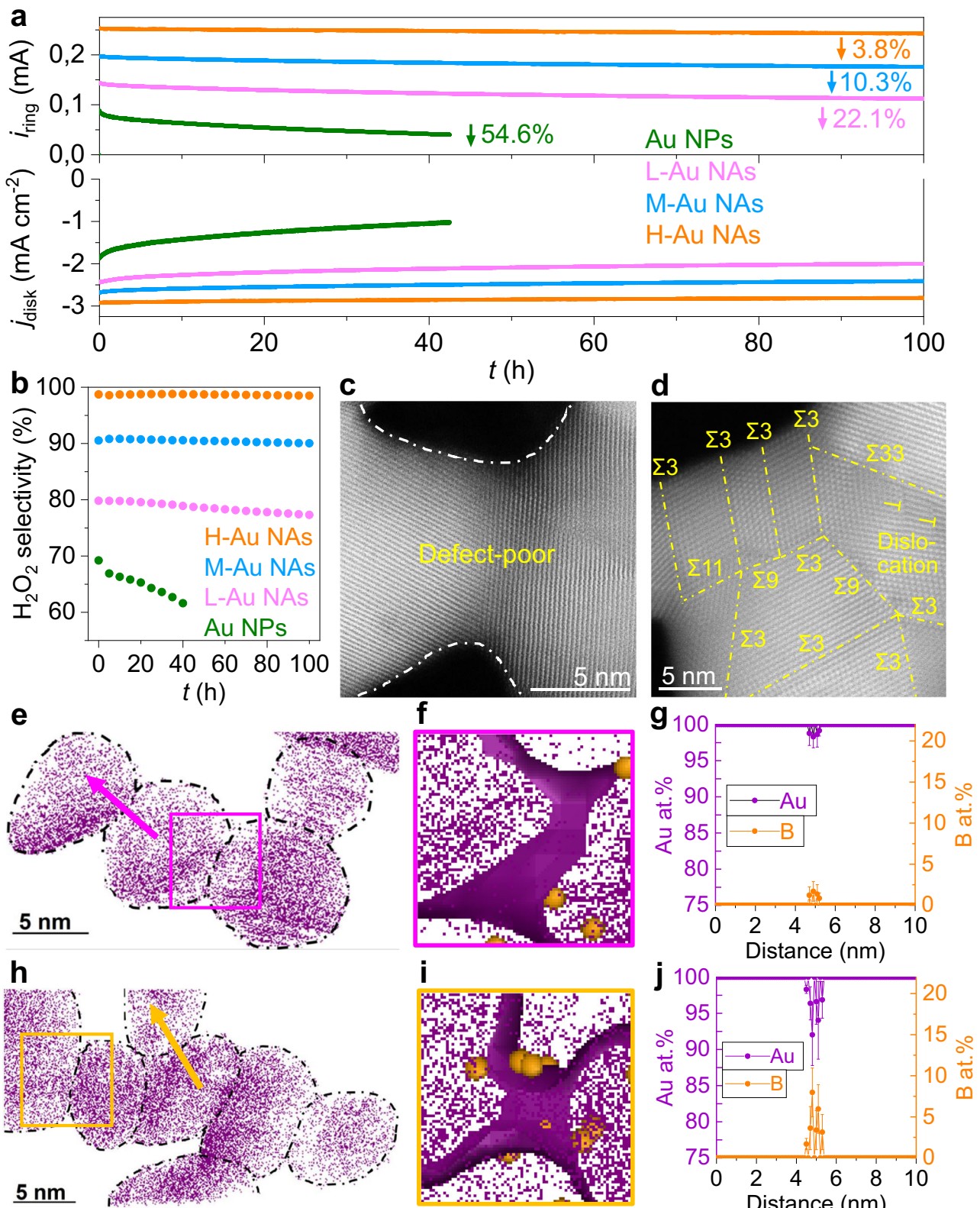

**Fig. 5 | Electrochemical and structural stability. a** Durability measurements of Au NPs, L-Au NAs, M-Au NAs and H-Au NAs at a fixed disk potential of 0.35 V. **b** Calculated $H_2O_2$ selectivity after durability tests. HAADF-STEM images of (**c**) L-Au NAs and (**d**) H-Au NAs after durability tests. A 2 nm thin-sliced tomogram from a 3D atom map (Supplementary Figs. S34, S37) of (**e**) L-Au NAs and (**h**) H-Au NAs (iso-composition surface >90 at.% Au). Extracted GB tomogram of (**f**) L-Au NAs and (**i**)

H-Au NAs of the region delineated by the box (isodensity surface of 150 Au atoms/ $nm^3$). The orange spheres represent B atoms. 1D compositional profiles of identified Au and B elements of (**g**) L-Au NAs and (**j**) H-Au NAs along the direction indicated by the arrow. Error bars indicate the standard deviation from a minimum of three independent measurements. Source data are provided as a Source Data file.

reducing their propensity for migration and annihilation. This finding underscores the necessity of adequate B concentration for maintaining structural stability and sustained ORR durability. While in this instance, we have limited our application to the two-electron ORR, it is noteworthy that the electronic effects resulting from the formation of GBs have the potential for optimizing the catalytic performance of our structures in the context of various other electrochemical reactions.

## Discussion

To summarize, we have synthetized Au nanoassemblies (NAs) with tunable grain boundary (GB) density. The kinetic driving force for NP collisions is provided by the sustained bubbling of $H_2$ gas, with the ability to control collision frequency and adjust the GB density of NAs by varying the gas flow rate. During the collision process and posterior re-orientation, defects (i.e., dislocations, stacking faults and atomic step ridges) and local strain in the vicinity of the GBs are induced. As the flow rate increases, so does the density of GBs, steps, dislocations, stacking faults as well as lattice expansion within the NAs. The combination of higher GB surface density, lower coordination number, other structural defects and lattice expansion enhances the catalytic performance of these Au NAs in more than two orders of magnitude (in specific and mass activity) with respect to GB-poor Au NPs for the two-electron ORR, surpassing the performance of other state-of-the-art catalysts in acidic media reported in the literature. The experimental assessments of the d-band center, coupled with theoretical computations, consistently validate that lattice expansion, optimized coordination number and GBs synergistically contribute to the enhancement of the two-electron ORR performance by fine-tuning the adsorption strength of the reaction intermediate *OOH on the surface of Au. The durability of the NAs is also improved with respect to the NPs, with negligible activity decay observed over 100 hours of operation. Notably, there exists a positive correlation between the structural stability of Au NAs, their corresponding two-electron ORR durability and the presence of segregated boron (B) species at their GBs. Segregation of B species at the GBs was observed, with B concentration increasing as the flow rates of gas bubbling increase, indicating the kinetic trapping of B at GBs. The improved durability of Au NAs can be attributed to the high concentration of B species segregated at the GBs, which inhibits decohesion or sliding of GBs, thereby stabilizing the defect-rich structure.

Our study presents a viable approach for modulating GB density, resulting in the development of an effective two-electron ORR catalyst in acidic media, notable for its high activity, selectivity, and stability. More significantly, our findings highlight the potential of GB engineering as a strategic tool for tailoring the properties of heterogeneous catalysts, paving the way for broader applications in catalyst design and optimization.

## Methods

### Material preparation

Synthesis of Au NPs. Initially, oleylamine (Sigma Aldrich) and oleic acid (Sigma Aldrich) were employed as the capping agents in the synthesis of Au NPs of ~10 nm. Subsequently, citrate (Sigma Aldrich) was employed to substitute oleylamine/oleic acid on the surface of the Au NPs. More synthesis and ligand exchange details are reported in Supplementary Materials.

Preparation of Au NAs. First, we employed a dialysis method to eliminate impurity ions from the citrate-capped Au NP solution. Specifically, the Au NP solution was placed inside dialysis tubing, which was immersed in a 1 L beaker containing DI water. Simultaneously, magnetic stirring was applied to expedite the diffusion of impurity ions away from the NP solution. The dialysis process was carried out continuously for three days, with DI water refreshed twice daily.

Next, we introduced high-purity $H_2$ gas into a 10 mL solution of citrate-capped Au NPs, with a concentration of ~1.6 μM, and maintained this bubbling process for 10 h. To investigate the influence of $H_2$ gas flow rate on the assembly of Au NPs, we employed three distinct flow rates, denoted as low (30 sccm), medium (100 sccm), and high (300 sccm), resulting in the formation of Au NAs named L-Au NAs, M-Au NAs, and H-Au NAs, respectively. The completion of Au NAs assembly was indicated by a change in the color of the Au NP solution from dark brownish-red to transparent. After discontinuing the $H_2$ gas bubbling, the solution was allowed to stand for 10 h, during which the Au NAs sedimented at the bottom of the beaker. Subsequently, careful removal of the supernatant was performed using a pipette, followed by thorough rinsing of the Au NAs with DI water.

### Microstructural characterizations

HAADF-STEM images were taken using a Thermo Fisher Titan microscope with probe Cs correction at an accelerating voltage of 300 kV, while HR-TEM was performed utilizing an Cs image aberration-corrected Thermo Fischer Titan Themis 60–300 microscope, also operated at 300 kV. 4D-STEM was used for performing grain orientation mapping and strain analysis of the nanoassemblies. The data was collected in the same Cs-probe corrected Thermo Fisher Titan microscope at 300 kV using the pixelated detector EMPAD. For the acquisition, a camera length of 940 mm and a probe convergence semiangle of 0.65 mrad were used. The detector pixel size was calibrated using a sample with (unstrained) Au NPs under the same conditions. XRD patterns were recorded on a Bruker Powder X-ray diffractometer equipped with a Cu Kα radiation source. XPS data were collected from an Al Kα X-ray Photoelectron Spectrometer (Thermo Scientific). XAS measurements were conducted at the BL14W1 beamline of the Shanghai Synchrotron Radiation Facility. The synchrotron storage ring operated at an energy of 3.5 GeV, while the linear electron accelerator operated at 150 MeV. APT tests were carried out using the LEAP 5076 XS instrument (Cameca) in pulse laser mode, with experimental parameters set at a temperature of 50 K, a detection rate of 1%, a laser energy of 60 pJ, and a laser pulse frequency of 125 kHz. The acquired data were then analyzed and reconstructed using standard voltage reconstruction protocols with the assistance of the commercially available IVAS 3.8.4 software. The zeta potential of the Au NP solution loaded into quartz cuvettes was determined using a Zetasizer Nano-ZS instrument (Malvern Instruments, UK) in high-resolution mode. Fourier-transform infrared spectroscopy (FTIR) spectra were acquired using a Bruker Tensor 27 FTIR spectrophotometer. The Au concentration in the ink was quantified via inductively coupled plasma atomic emission spectroscopy (ICP-AES) using a 710-ES instrument from Varian. More characterization and analysis details are reported in Supplementary Materials.

### Electrochemical measurements

The electrochemical measurements were carried out using a CHI760E potentiostat (CHI Instrument) using 0.1 M $HClO_4$ aqueous solution as the electrolyte. The rotating ring disk electrode (RRDE) measurements were run at 25 °C in a typical three-electrode cell. A platinum (Pt) foil (99.99%, Sigma Aldrich) and an Ag/AgCl electrode (3.0 M KCl, CH Instrument) were used as the counter and reference electrode, respectively. All the potentials were converted to reversible hydrogen electrode (RHE). The RHE calibration of Ag/AgCl reference electrode in 3.0 M KCl was performed in a high purity of $H_2$ saturated 0.1 M $HClO_4$ solution where polished Pt wires were used as the working and counter electrodes. The linear scanning voltammetry (LSV) was run at a scan rate of 1 mV/s, and the potential of zero current was taken as the reaction potential of the hydrogen electrode. The potential measured with the Ag/AgCl electrode in 3.0 M

KCl were calculated with Eq. (1):

$$E_{RHE} = E_{Ag/AgCl} + 0.279 V \qquad (1)$$

A RRDE assembly (PINE Instrument) consisting of a glassy carbon rotating disk electrode ($\Phi$ = 5.6 mm, geometric area = 0.247 cm$^2$) and a Pt ring ($\Phi$ = 15.0 mm) was used, with a theoretical collection efficiency of 40%. Experimentally, the apparent collection efficiency (N) was determined to be 36% in the ferrocyanide/ferricyanide half reaction system at a rotation rate between 400 and 1600 rpm.

To prepare the ink, 0.3 mg of as-synthesized Au catalysts were dispersed in 495 µL DI water by sonicating for 1 h, followed by adding commercial Vulcan carbon (XC-72) with a mass four times than Au samples and sonicating for another 1 h to make Au catalysts/C. Then, the Au catalysts/C were dispersed in the mixture of 495 µL isopropanol and 10 µL 5% Nafion solution to from a homogeneous ink by sonicating for 1 h. The concentration of Au was controlled to be 0.3 mg$_{Au}$/mL measured by ICP-AES. The uniform catalyst layer was prepared by pipetted 5 µL of the ink onto the disc electrode without obvious pin holes or uncovered edge followed by vacuum drying at ambient conditions. The loading amount of Au was controlled at 6.1 µg/cm$^2$.

Before the activity test, both glassy carbon rotating disk electrode with Au catalysts and ring electrode were cleaned by cyclic voltammetry (CV) for 500 cycles in N$_2$-saturated 0.1 M HClO$_4$ electrolyte solution until a stable CV curve was obtained. The electrochemical active surface area (ECSA) was calculated by integrating the charge for the reduction of gold oxide on CV curves in the range of 0.03–1.6 V at 100 mV/s by assuming a value of 390 µC/cm$^2$. The H$_2$O$_2$ activity and selectivity were measured from polarization curves in O$_2$-saturated 0.1 M HClO$_4$ electrolyte solution between 0.8 and 1.4 V with a scan rate of 10 mV/s with rotating rate of 1600 rpm, while holding the potential of the Pt ring electrode at 1.2 V. The ORR current was corrected by subtracting the current obtained in an N$_2$-saturated electrolyte from that measured in O$_2$-saturated conditions. All the measured ring currents were also corrected using the collection efficiency, N, of RRDE set-up to obtain the overall current density as all the H$_2$O$_2$ generated was detected. The H$_2$O$_2$ selectivity was calculated using Eq. (2):

$$H_2O_2 (\%) = 200 \times \frac{I_{ring}/N}{I_{disk} + I_{ring}/N} \qquad (2)$$

where $I_{ring}$ is the ring current, $I_{disk}$ is the disk current, and N is the collection efficiency.

The long-term stability tests for Au catalysts were conducted in the RRDE set-up (chronoamperometry at 0.35 V vs. RHE, rotation speed of 1600 rpm).

## Data availability
All data needed to evaluate the conclusions are presented in the paper and the Supplementary Materials. Source data are provided with this paper.

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

## Acknowledgements

X.G. acknowledges financial support from the MPG scholarship. B.G. is grateful for funding from the ERC for the project SHINE (ERC-CoG) #771602, and to the DFG for funding through the Leibniz Award. The authors thank Uwe Tezins, Christian Broß and Andreas Sturm for their support at the FIB and APT facilities at MPIE. The authors thank Dr. Xuyang Zhou for valuable discussions on the 4D-STEM data analysis. Work at the Molecular Foundry was supported by the Office of Science, Office of Basic Energy Sciences, of the U.S. Department of Energy under Contract No. DE-AC02-05CH11231.

## Author contributions

X.G. and M.V.P. contributed equally to this work. X.G. and B.G. designed the overall experiment. X.G. performed the material preparation, electrochemistry, FIB and APT measurements. M.V.P. performed the HAADF-STEM and HRTEM experiments under the supervision of C.S. M.V.P. performed the 4D-STEM experiments with the support of C.O., S.Z., and C.H.L. X.G. conducted the APT analysis with assistance from Y.M. and B.G. Z.W., P.L., and X.G. conducted the theoretical simulation. X.G., M.V.P., P.L., Y.M. C.S., and B.G. collectively wrote the manuscript. All authors contributed and have given approval to the final version of the manuscript.

## Funding

## Competing interests

The authors declare no competing interests.
