## [Peer Review File · Nature Communications]

Grain Boundary Engineering for Efficient and Durable ElectrocatalysisREVIEWER COMMENTS

Reviewer #1 (Remarks to the Author):

In this paper, the authors introduced the GB density controlled synthesis of Au NAs through modifying the collision frequency. The study also showed advanced 4D STEM technique to measure GB density in the nanostructures. However, some of synthesis and echem sections need major revisions to meet the standard of the Nat. Comm. Here are comments;

1. Can Ar or N₂ gases play a similar role of H₂ during synthesis, or are there specific reasons for selecting H₂? Furthermore, considering the introduction of H₂ gas into the Au samples, it is essential to address any potential formation of AuH and its influence on electrocatalytic activity.
2. The prepared sample showed high porosity and GB density. Which of the following structural characteristics has a greater effect on electrocatalytic activity? A description of the major factor affecting electrocatalysis is needed.
3. The abstract began by discussing CO₂RR, which clouds the main claim of the study. The authors should revise the abstract to align it with the ORR research in the main text.
4. Considering that a coordination number of 8 is optimal for GB-rich environments, it would be valuable to ascertain the coordination number of the catalyst exhibiting the best ORR performance.
5. While the manuscript discussed the relationship between GB density and ORR, it is essential to provide more clarity regarding the specific ORR catalysis targeted in the introduction. Previous studies, such as those in Nano Research (2020, 13, 3310-3314) and ACS Catal (2022, 12, 3516-3523), have explored the relationship between GB density and ORR

Reviewer #2 (Remarks to the Author):

The manuscript presents a novel controllable method for adjusting grain boundary densities in gold nano-assemblies. It investigates the relationship between grain boundary density and the performance of the two-electron oxygen reduction reaction (ORR). Notably, the developed synthesis method allows precise grain boundary density adjustments without altering grain size, providing an improved control experiment for studying the impact of grain boundaries on catalytic activity. However, despite these advancements, the manuscript lacks new insights regarding the correlation between grain boundaries and reaction outcomes. Additionally, certain aspects of the article remain unclear and require further clarification. Therefore, I recommend a major revision of the manuscript before considering it for publication. The detailed comments are as follows:

1. In the introduction section, the manuscript would benefit from citing more recent works to emphasize its significance. Specifically, the importance of ORR selectivity towards the two-

electron pathway should be highlighted (Materials Today Catalysis 4 (2024): 100038, Advanced Science 8.15 (2021): 2100076). The importance of advanced characterization methods for catalytic research (Materials Today Catalysis (2023): 100033) also needs to be stressed, as this manuscript relies on advanced characterization techniques.

2. The concept that grain boundaries contribute to catalytic activity is not new, and similar works have been extensively reported (Science 373.6562 (2021): 1518-1523, Nature Communications 11.1 (2020): 57). The authors need to illustrate the importance of their work compared to other already published papers.

3. Detailed explanations of the Au nano-assembly synthesis mechanism are crucial. The authors need to address why Au nanoparticles coalesce spontaneously upon collision and illustrate the role of hydrogen gas bubbles in detail. If the bubbles only provide external forces, can other gases or methods (mechanical stirring, ultrasound) be used to achieve the same effect?

4. The authors need to provide more characteristic results for the original Au NPs to demonstrate the grain boundary density before assembly.

5. In Fig. 1d, the Au NAs displayed many fivefold twin structures. According to Science 367.6473 (2020): 40-45, there should be two formation mechanisms for the fivefold twin structure. The authors should analyze the formation mechanism of Au NAs accordingly.

6. In Figs. 2f-h, the authors analyzed the strain of individual atoms by fitting 2D Gaussians to find the atomic column position. This result should be more reliable for a single phase (inside a single grain) since the atomic spacing at different phases and at the interface is incomparable. The method here should be further validated.

7. In Figs. 2j and k, there seems to be a stronger strain along the yy direction compared to the xx direction. This requires further explanation.

8. Regarding the lattice expansion induced by the grain boundary, the authors claim that "the lattice expansion derived from EXAFS analysis of L-Au NAs, M-Au NAs, and H-Au NAs are approximately 6%, 4%, and 2%, respectively." The discrepancy between lattice expansion and grain boundary density (H-Au NAs inducing the lowest expansion despite the highest grain boundary density) should be addressed.

9. In Fig. 4b, why does the selectivity of Au NPs change with potential while Au NAs maintains the same selectivity? The authors should provide deeper discussion to demonstrate the superiority of the developed Au NAs catalysts.

10. What are the grain boundary (GB) type distributions of L-Au NAs and M-Au NAs? Will the rate of hydrogen gas bubbles affect the GB type, and what is the impact of GB types on ORR performance?

11. For the stability concerning the B species, the authors claimed that "After the durability testing,

the concentration of B at the GBs of H-Au NAs remains nearly constant, whereas in L-Au NAs, B is nearly gone. This observation further underlines that the concentration of B species at GBs is a primary contributor to the superior stability exhibited by H-Au NAs." However, the change in B content seems more likely to be a result rather than a cause. In other words, it should be the instability of the sample that leads to the loss of B, rather than the loss of B causing a decrease in stability. Therefore, there is still no direct evidence to suggest a direct association between B species and sample stability. Furthermore, the impact of B species on stability seems to deviate from the main theme of the article, which focuses on grain boundary engineering.

Reviewer #3 (Remarks to the Author):

The paper entitled « Grain Boundary Engineering for Efficient and Durable Electrocatalysis » submitted to Nature Communication focuses on the beneficial role on grain boundaries (GB) in the surface adsorption properties of Au nanoparticles. The authors successfully tuned the density of GB while maintaining a similar size of the coherent domains of Au nanoparticles. This work is of high interest to the electrocatalysis community. I recommend publication after addressing the following comments in a revised version of the manuscript. By doing so, the conclusions will certainly be strengthened.

1- My Main concern relates to the role of boron in activity/stability relationships. First, boron is only mentioned at the end of the abstract "Their structural and catalytic stability can be attributed to segregated boron at the GBs..." without any word of its origin. Only at pages 9/15, it is learnt that boron comes from boric acid used in the nanoparticle's synthesis process. Furthermore, quantitatively speaking, "traces amounts of boron" are mentioned, which I find misleading for the reader. In Figure 5 and S25, the 3D atom mapping shows a boron content of around 5-10 at% located in the vicinity of the GBs. Such content is substantial! The authors should clarify the role of boron because it likely also influences the reactivity of the Au nanoparticles by modifying the electronic structure of Au.

2-Following up on my initial comment, it would have been interesting to disentangle the roles of boron and structural defects. In the case of "classical" Au nanoparticles, boron is located at the surface, whereas for the NAs Au with no boron detected on the surface but at the vicinity of the GBs, the comparison of activity and stability trends is more complicated. Atom probe tomography results indicate a clear relationship between boron content and oxygen reduction activity. Since boron content and GB density are linked, it is unclear if the authors can claim that GBs (and structural defects) enhance activity while the presence of boron improves the stability of GBs. Can the authors synthesize Au nanoparticles without using boric acid or by replacing it with another type of acid?

3- The role of structural defects in electrocatalysis of the oxygen reduction reaction has been studied by L. Dubau and F. Maillard. I recommend that the following publications be mentioned in this manuscript: Nat. Mater. 17 (2018) 827-833, ACS Catal., 7 (2017) 3072-3081, and ACS Catal., 6 (2016) 4673-4684.

Response to Reviewers' Comments

Ms. ID: NCOMMS-24-15705-T

Title: Grain Boundary Engineering for Efficient and Durable Electrocatalysis

Reviewers' comments in normal type

Author response in italics

Verbatim quotes from the revised text are in boldface and highlighted in yellow

• Reviewer 1

Comments:

In this paper, the authors introduced the GB density controlled synthesis of Au NAs through modifying the collision frequency. The study also showed advanced 4D STEM technique to measure GB density in the nanostructures. However, some of synthesis and echem sections need major revisions to meet the standard of the Nat. Comm. Here are comments;

Author reply: We thank the reviewer for the positive opinion about our work.

1. Can Ar or N₂ gases play a similar role of H₂ during synthesis, or are there specific reasons for selecting H₂? Furthermore, considering the introduction of H₂ gas into the Au samples, it is essential to address any potential formation of AuH and its influence on electrocatalytic activity.

*Author reply: We would like to thank the reviewer for raising this point. To investigate the role of H₂ gas bubbling in the self-assembly of Au NPs, we bubbled Ar gas and N₂ gas instead of H₂ into Au NP solutions in a mildly alkaline environment (pH=10), maintaining the rest of the conditions unaltered. After 10 h of Ar gas purging, most Au NPs remained separated (**Figure R1a**), indicating that Ar gas purging is insufficient for assembling NPs into a network. FTIR analysis showed that all the citrate ligands were removed from the Au NP surface due to the destabilization of citrate ligands in the mildly alkaline environment (**Figure R1b**).¹ Despite the absence of steric repulsion from citrate ligands, the NPs still failed to assemble into a network, suggesting the presence of other inhibitory factors. ζ -potential measurements revealed that the ζ -potential of the Au NP solution remained at -33 mV after 10 h of Ar gas purging (**Figure R1c**), indicating that the Au NP surfaces were still negatively charged, likely due to surface-bound OH⁻. This electrostatic repulsion between negatively charged NPs may explain the failure of NP assembly with Ar gas purging.² Similarly, N₂ gas purging also did not facilitate NP network assembly (**Figure R1d-R1f**).*

*In contrast, H₂ gas purging successfully assembled NPs into a network (**Figure R1g**). FTIR analysis confirmed the complete removal of citrate ligands from the NP surface (**Figure R1h**), and ζ -potential measurements showed that the ζ -potential was nearly zero (**Figure R1i**). This suggests that the NP surface charge shifted from negative to nearly neutral, likely due to competitive adsorption between OH⁻ and hydrogen atoms. As hydrogen atoms gradually occupied the binding sites previously held by OH⁻, the NP surface became neutral, eliminating electrostatic repulsion.*

The inability of inert gases like Ar and N₂ to adsorb onto the NP surface, thus maintaining a negatively charged surface, contrasts with the effects of H₂ gas. This finding highlights the dual role of hydrogen gas bubbling in the formation of Au nanoparticle assemblies. Firstly, excess hydrogen gas introduces competitive adsorption between OH⁻ and hydrogen, displacing negatively charged ligands and transforming the NP surface to a neutral state, thereby removing electrostatic repulsion. Secondly, the physical agitation from H₂ gas bubbling generates turbulence and convection currents, increasing the frequency and energy of NP collisions. When NPs collide, they can coalesce at regions lacking citrate ligands, forming GBs. This spontaneous attachment is driven by the reduction of the system's total energy, with coalescence eliminating high-energy surface faces and minimizing overall energy.³ Structural defects such as step sites, stacking faults, and dislocations can form if NPs are not perfectly aligned during attachment, typically leading to anisotropic growth and nanowire formation.⁴

*Regarding the question on the potential formation of AuH. To date, limited characterization methods are available for detecting gold hydrides, with Raman spectroscopy being the most commonly employed.⁵ Gold hydrides exhibit two strong bands at 460 and 910 cm⁻¹, along with weaker bands at 815, 1091, 1125, and 2100 cm⁻¹, which are absent in pure gold.⁵ Raman spectra of H-Au NAs show no discernible peaks at 460 and 910 cm⁻¹, indicating the absence of gold hydrides in H-Au NAs (**Figure R2**). This may be due to the weak adsorption of hydrogen on the Au surface,⁶ leading to their facile desorption. During the attachment of Au NPs to form GBs, hydrogen likely desorb from the surface, preventing the formation of gold hydrides. This is plausible as known solid gold hydrides (AuH, AuH₂) are unstable and can only be synthesized under high temperature and pressure conditions.⁷*

Figure R1. (a) TEM image of the NPs obtained by purging Ar gas into a citrate-capped Au NP solution at pH 10 for 10 h. (b) FTIR spectrum of the Au NP solution at pH 10 subjected to Ar gas purging for 0 h and 10 h, respectively. (c) Time-dependent variation of the ζ -potential for citrate-capped Au NP solution at pH 10 during Ar gas purging. TEM image (d), FTIR spectrum (e) and time-dependent variation of ζ -potential (f) for citrate-capped Au NP solution at pH 10 during N₂ gas purging. TEM image (g), FTIR spectrum (h) and time-dependent variation of ζ -potential (i) for citrate-capped Au NP solution at pH 10 during H₂ gas purging.

Figure R2. Raman scattering spectra of H-Au NAs and the AuH from literature⁵. Notice how H-Au NAs is missing the bands at 460 and 910 cm^{-1} , characteristic of gold hydrides.

We added **Figure R1** as **Figure S6** to the supplementary information, and also added the following sentences to the main text.

Page 3: “The assembly of Au NPs comprises three distinct stages. Firstly, under mildly alkaline conditions (pH=10), the initially capped citrate ligands undergo gradual detachment from the NP surface (fig. S4) due to their instability (fig. S5), and as a consequence, OH^- ions can adsorb onto the NP surface. During this stage, the steric repulsion between Au NPs originating from the citrate ligands is alleviated, although electrostatic repulsion persists due to the adsorption of OH^- ions (fig. S6a-c). Secondly, the continuous bubbling of an excess amount of hydrogen gas into the Au NP solution facilitates competitive adsorption between OH^- ions and H atoms. This results in the gradual replacement of surface-bound OH^- ions by H atoms, thereby transforming the negatively charged NP surface into a nearly neutral state and eliminating electrostatic repulsion between Au NPs. Thirdly, the introduction of H_2 gas into the NP solution induces turbulence and convection currents, thereby increasing the frequency of NP collisions. Upon approaching each other, two NPs can undergo spatial rotation, followed by attachment and coalescence at regions devoid of ligands to form a GB. The primary driving force for this spontaneous attachment is the elimination of bare regions lacking ligand protection, possessing high surface energies, thereby reducing the

total energy of the system. Other structural defects such as step sites, stacking faults, and dislocations may arise due to imperfect orientation during the attachment process. Furthermore, NP attachment typically results in anisotropic growth, giving rise to nanowire morphology. If other gases (e.g., Ar, N₂) are used instead of H₂, the NP attachment does not occur (fig S6di) due to the electrostatic repulsions between OH⁻ capped NPs' surfaces.”

2. The prepared sample showed high porosity and GB density. Which of the following structural characteristics has a greater effect on electrocatalytic activity? A description of the major factor affecting electrocatalysis is needed.

Author reply: We thank the reviewer for the good suggestion. We employed Brunauer-Emmett-Teller (BET) nitrogen adsorption/desorption experiments to characterize the specific surface area and pore structure of the different Au NAs.

L-Au NAs, M-Au NAs, and H-Au NAs all demonstrated large specific surface areas and high porosity (Figure R3, Table R1). Among these, L-Au NAs exhibited the largest specific surface area, pore diameter, and pore volume, while H-Au NAs had the smallest specific surface area, pore diameter, and pore volume. The specific surface area and porosity of M-Au NAs were intermediate between those of L-Au NAs and H-Au NAs. The trends in BET-measured specific surface areas of Au NAs were consistent with those measured electrochemically. When increasing H₂ flow rate, the collision frequency between Au NPs building blocks increases, resulting in lower surface area and higher GB density.

As the GB density increases, the porosity of Au NAs decreases and their two-electron oxygen reduction activity and selectivity improve. This indicates that high porosity is not the source of two-electron ORR catalytic activity; rather, high GB density is the determining factor.

Figure R3. Nitrogen adsorption–desorption isotherms of L-Au NAs (a), M-Au NAs (c) and H-Au NAs (e). Barrett–Joyner–Halenda pore size distribution plots of L-Au NAs (b), M-Au NAs (d) and H-Au NAs (f).

Table R1. BET results of L-Au NAs, M-Au NAs, and H-Au NAs.

Sample	BET surface area (m ² /g)	Pore diameter (nm)	Pore volume (cm ³ /g)
L-Au NAs	25.2	23.6	0.19
M-Au NAs	22.8	17.6	0.15
H-Au NAs	19.5	11.2	0.10

We added **Figure R3** as **Figure S18**, **Table R1** as **Table S1** to the supplementary information, and also added the following sentences to the main text.

Page 6: “Brunauer-Emmett-Teller (BET) measurements indicate that L-Au NAs, M-Au NAs, and H-Au NAs possess large specific surface areas and high porosity (fig. S18, table S1). Among these, L-Au NAs exhibits the largest specific surface area, pore diameter, and pore volume, while H-Au NAs has the smallest. The specific surface area and porosity of M-Au NAs are intermediate between those of L-Au NAs and H-Au NAs. The trends in BET-measured specific surface areas of Au NAs are consistent with electrochemical measurements.”

3. The abstract began by discussing CO₂RR, which clouds the main claim of the study. The authors should revise the abstract to align it with the ORR research in the main text.

Author reply: We thank the reviewer for highlighting this important point. The abstract has been revised to align it better with the relationship between GBs and ORR catalysis.

To clarify this point, the abstract has been rewritten the abstract:

Page 1: Grain boundaries (GBs) are two-dimensional defects that can increase the catalytic activity of noble metal catalysts for key electrochemical reactions such as the oxygen reduction reaction (ORR). However, previous strategies for modifying the GB density of electrocatalysts result in simultaneous changes in particle size, shape and morphology, prompting uncertainty about the specific role of GBs in boosting catalytic activity. This uncertainty is further driven by the incomplete characterization of GB type distribution, their atomic arrangement and local composition, hindering the understanding of GBs’ structure-catalytic activity relationship.

Furthermore, stabilizing these active defects during electrocatalysis represents a considerable challenge, as factors such as heating or an electric field can induce atomic mobility at the GBs.

4. Considering that a coordination number of 8 is optimal for GB-rich environments, it would be valuable to ascertain the coordination number of the catalyst exhibiting the best ORR performance.

*Author reply: We agree that it is valuable to study the effect of coordination number (CN) on the two-electron ORR activity. We performed DFT simulations to elucidate the relationship between CN and the two-electron ORR activity. We constructed six atomic models with CNs of 6, 7, 8, and 9 on the Au (111) surface to simulate catalytic sites (Figure R4a-4b). The volcano plot was calculated using the limiting potential (U_L) as the performance indicator, defined as the maximum potential at which both the one-electron reduction of O_2 to $*OOH$ and the subsequent one-electron reduction of $*OOH$ to H_2O_2 are energetically favorable.*

*The model with a CN of 8 exhibits a free energy deviation of merely -0.01 eV from the ideal state, placing the $*OOH$ adsorption energy at the volcano plot's peak (Figure R4c,d). Increasing the CN to 9 is detrimental for two-electron ORR performance. At CN=9, the $*OOH$ adsorption energy lies on the right leg of the volcano plot (Figure R4c), indicating a low affinity for $*OOH$ adsorption, which hinders the O_2 adsorption and slows down the reaction (Figure R4d). Similarly, reducing the CN to more than 8 is also detrimental for catalytic performance. In these cases, the $*OOH$ adsorption energy falls on the left leg of the volcano plot, indicating a too strong affinity for $*OOH$ adsorption, which makes it difficult for the product to desorb, thereby hindering subsequent reactions and leading to catalyst poisoning. DFT calculations suggest that a CN of ~8 offers optimal two-electron ORR activity, contributing to the superior ORR activity and selectivity observed in H-Au NAs.*

Figure R4. (a) Slab model of the Au (111) surface, highlighting the surface atom used for OOH binding energy calculations in blue. (b) Surface atom (blue) used for OOH binding energy calculations with different coordination numbers (CNs) on the Au (111) surface, showing only the nearest neighbor atoms. (c) Calculated ORR activity volcano relationship between the limiting potential (U_L) and the free energy of $*OOH$ (ΔG_{*OOH}) for the two-electron pathway to H_2O_2 . (d) Calculated reaction coordinate diagrams for Au with different CNs.

We added **Figure R4** as **Figure S21** in the supplementary information, and also added the following sentences to the main text.

Page 9: “The coordination number also significantly influences two-electron ORR performance (fig. S21). When it has a value of 9 or larger, the $*OOH$ adsorption energy is too weak (located on the right side of the volcano plot, fig. S21c), reflecting a diminished affinity for $*OOH$ and

hindering O₂ adsorption. Reducing the coordination number enhances *OOH adsorption strength, progressively shifting it towards the volcano plot's peak (fig. S21c). A coordination number of 8 achieves a free energy deviation of just -0.01 eV from the ideal state, positioning *OOH adsorption energy nearly at the plot's peak. If the coordination number is further decreased, the affinity towards *OOH is too high, which can hinder product desorption and cause catalyst poisoning, thereby slowing subsequent reactions. Thus, a coordination number of approximately 8 is optimal for achieving superior two-electron ORR activity, highlighting its role in optimizing both activity and selectivity.”

Page 12: “The experimental assessments of the d-band center, coupled with theoretical computations, consistently validate that lattice expansion, optimized coordination number and GBs synergistically contribute to the enhancement of the two-electron ORR performance by fine-tuning the adsorption strength of the reaction intermediate *OOH on the surface of Au.”

5. While the manuscript discussed the relationship between GB density and ORR, it is essential to provide more clarity regarding the specific ORR catalysis targeted in the introduction. Previous studies, such as those in Nano Research (2020, 13, 3310-3314) and ACS Catal (2022, 12, 3516-3523), have explored the relationship between GB density and ORR.

Author reply: We thank the reviewer for raising this point. It is indeed essential to clarify more about ORR catalysis in the Introduction. Several studies have demonstrated that Pt or Pt-based alloy nanomaterials rich in GBs often exhibit high four-electron ORR catalytic activities.⁸⁻¹² For example, Huang et al. developed GB-rich Pt nanowire and nanoplate,⁸ which showed high four-electron ORR activity due to the increased residence number and extended residence time of oxygen on surface GB sites. These studies are promising, however, they have certain limitations and there remains room for further improvement: 1) the nanomaterials used for ORR activity comparisons differ not only in GB density but also in size, shape, and morphology. This raises questions about whether the observed enhancement in ORR activity is genuinely attributable to GBs. 2) the GB density in these studies is not experimentally measured but rather derived from simulations, casting doubt on the quantitative reliability of the reported GB densities. 3) precise atomic arrangements, local composition, and GB types were not thoroughly characterized, making it challenging to elucidate the relationship between GB structure and catalytic activity. 4) no methods were proposed to controllably adjust GB density

without altering other structural features (e.g., size, composition), making it difficult to precisely regulate the number of these active sites. 5) mechanistic studies on how GBs enhance ORR activity are limited. Aside from Huang et al.'s viewpoint that high ORR activity is due to a larger residence number and longer residence time of oxygen on GB sites,⁸ there is a lack of atomic-level understanding from other studies. 6) current research on GB-regulated ORR activity focuses on the four-electron pathway. The potential of using GBs to regulate catalyst selectivity (e.g., two-electron ORR) has not been explored.

Additionally, it is notable that there is no consensus in the literature on the stability of GBs in ORR. For instance, some studies suggest that Pt GBs are stable during ORR,⁸ while Choi et al. found that polycrystalline PtCo nanowires exhibited significantly lower ORR stability compared to single-crystal PtCo nanowires.⁹ Atoms at GBs are thermodynamically unstable during ORR due to their high energy, which significantly alters the initial structure and ultimately deactivates the catalyst. Thus, enhancing the stability of atoms at GBs during ORR is highly challenging. The inconsistent reports on the stability of GBs in ORR may be attributed to variations in local composition at GBs. Trace impurities, which are difficult to avoid during material synthesis, can thermodynamically accumulate at GBs.¹³ These impurities, often light elements at low concentrations, are challenging to detect using macroscopic spectroscopic techniques such as XPS.¹⁴ Additionally, these impurities are usually embedded within GBs, rendering many surface analysis methods ineffective. Therefore, atomic-scale, spatially-resolved characterization techniques, such as atom probe tomography, are essential for studying the local composition distribution near GBs.

To clarify this point, we have rewritten the introduction to include previous studies that explored the relationship between GBs and the four-electron ORR activity.

Page 1-2: “Grain boundaries (GBs) are two-dimensional structural defects that can effectively synergize these strategies: they can induce local lattice strain, atomic step ridges, and offer unique active sites for reactions like the four-electron oxygen reduction reaction (ORR). For instance, Maillard et al. demonstrated that Pt–Ni nanocrystals with increased surface distortion exhibit superior four-electron ORR activities compared to their less defective counterparts. However, due to the challenge of introducing GBs in nanocatalysts in a controlled manner, the understanding of how GBs modulate catalytic activity and their potential to catalyze other reactions remains limited. Existing strategies for tuning GB density modify also the size, shape and morphology of

nanocatalysts, hampering the interpretation of the relationship between GBs and catalytic activity. Our understanding is also limited by incomplete characterization of GB type distribution, their atomic arrangement and local composition. This lack of understanding leads to a negligence of the potential of GBs to regulate not only the catalytic activity but also selectivity towards certain pathways. For instance, having an active catalyst for oxygen reduction that is also selective towards its two-electron pathway would enable the electrochemical production of hydrogen peroxide (H_2O_2), a chemical widely used in various industries. In addition, the stability of GBs during electrocatalysis is also debated. Some studies suggest that Pt GBs are stable during four-electron ORR,⁸⁻¹² while others report poor stability of GBs in polycrystalline PtCo nanowires compared to single-crystal counterparts.⁹ Atoms at GBs are thermodynamically unstable due to their high energy, which can lead to structural degradation and catalyst deactivation. Inconsistent stability reports may be due to local compositional variations and trace impurities at GBs, which are challenging to detect with conventional techniques but have been reported to effectively stabilize these defects.”

Reviewer 2

Comments:

The manuscript presents a novel controllable method for adjusting grain boundary densities in gold nano-assemblies. It investigates the relationship between grain boundary density and the performance of the two-electron oxygen reduction reaction (ORR). Notably, the developed synthesis method allows precise grain boundary density adjustments without altering grain size, providing an improved control experiment for studying the impact of grain boundaries on catalytic activity. However, despite these advancements, the manuscript lacks new insights regarding the correlation between grain boundaries and reaction outcomes. Additionally, certain aspects of the article remain unclear and require further clarification. Therefore, I recommend a major revision of the manuscript before considering it for publication. The detailed comments are as follows:

Author reply: We express our gratitude to the reviewer for the positive assessment of our research.

1. In the introduction section, the manuscript would benefit from citing more recent works to emphasize its significance. Specifically, the importance of ORR selectivity towards the two-electron pathway should be highlighted (Materials Today Catalysis 4 (2024): 100038, Advanced Science 8.15 (2021):

2100076). The importance of advanced characterization methods for catalytic research (Materials Today Catalysis (2023): 100033) also needs to be stressed, as this manuscript relies on advanced characterization techniques.

Author reply: We would like to thank the reviewer for raising these two points.

Regarding the importance of ORR selectivity towards the two-electron pathway we have added the following sentences to the Introduction to clarify this point.

Page 2: “This lack of understanding leads to a negligence of the potential of GBs to regulate not only the catalyst activity but also its selectivity towards a certain pathway. For instance, having an active catalyst for oxygen reduction, which is also selective towards its two-electron pathway, would enable the electrochemical production of hydrogen peroxide (H₂O₂), a chemical widely used in various industries.^{15-17”}

For better highlighting the importance of advanced characterization methods for catalytic research, the following sentences have been added to the Introduction with the respective references.

Page 2: “In addition, the stability of GBs during electrocatalysis is also debated. Some studies suggest that Pt GBs are stable during four-electron ORR,⁸ while others report poor stability of GBs in polycrystalline PtCo nanowires compared to single-crystal counterparts.⁹ Atoms at GBs are thermodynamically unstable due to their high energy, which can lead to structural alterations and catalyst deactivation.⁹ Inconsistent stability reports may be due to local compositional variations and trace impurities at GBs,¹³ which are challenging to detect with conventional techniques but have been reported to effectively stabilize these defects. For instance, boron (B) is reported to stabilize the GBs of bulk polycrystals by lowering their energy, enhancing overall structural integrity.^{18, 19} However, its effect on GB stabilization in nanocatalysts remains unexplored. Advanced atomic-scale characterization methods, such as atom probe tomography (APT),¹³ high-angle annular dark-field (HAADF)-scanning transmission electron microscopy (STEM),²⁰ and 4D-STEM,²¹ are essential for investigating the local composition and structure near GBs, thus advancing our understanding of their catalytic roles.”

2. The concept that grain boundaries contribute to catalytic activity is not new, and similar works have been extensively reported (Science 373.6562 (2021): 1518-1523, Nature Communications 11.1 (2020):

57). The authors need to illustrate the importance of their work compared to other already published papers.

Author reply: We would like to thank the reviewer for raising this point. Compared with previous studies demonstrating that GBs enhance catalytic activity, our work is significant for several reasons:

1) Controlled adjustment of GB density: Unlike prior methods that alter GB density along with other structural features such as grain size, we achieved controlled adjustment of GB density without changing other structural characteristics. Catalytic GB-rich materials have been synthesized using techniques like metal oxide reduction,²² steam treatment,²³ vapor deposition,²⁴ laser ablation,²⁵ and electrodeposition with additives²⁶. However, controlling GB density in nanomaterials during synthesis remains challenging, complicating the establishment of reliable structure-activity relationships. Thermal post-treatment, commonly used to vary GB density, often results in concurrent structural changes, such as grain size increase,²⁴ making it difficult to interpret GB density–activity correlations. In this work, we formed GB-rich Au NAs through NP surface collision, attachment, and coalescence without capping agents. NP collisions were driven by continuous H₂ gas bubbling, with the gas flow rate controlling the collision frequency and hence the GB density. This method allows precise tailoring of GB density in GB-rich nanomaterials, which was not possible with the previously existing methods.

2) Understanding mechanisms of GB-enhanced catalytic activity: Most studies attribute the enhanced catalytic activity of GB-rich nanomaterials to strain fields generated near GBs.^{23, 25, 27} However, theoretical calculations suggest that strain fields alone cannot account for the significant activity increase.²³ Structural defects such as step atoms and dislocations near GBs have also been proposed as contributing factors,^{28, 29} but these defects are difficult to quantify. Thus, the exact mechanisms by which GBs enhance catalytic activity are not fully understood, limiting the rational design and optimization of GB-rich catalysts. This work shows that changes in coordination number, reflecting structural defects, are key to understanding GB-enhanced activity. The coordination number of GB-rich Au NAs decreases with more structural defects, optimizing the binding energy of critical reaction intermediates like OOH and enhancing two-electron ORR activity and selectivity. The synergistic effects of increased strain and decreased coordination number amplify catalytic performance beyond what strain fields alone can achieve.

3) Expanded catalytic selectivity of GBs: GBs in different materials have been identified to improve the catalytic activity for specific reactions: Pd GBs for methane activation,²³ Pt GBs for four-electron ORR,⁸ Ir GBs for oxygen evolution reaction,³⁰ Cu GBs for CO₂ reduction reaction,³¹ and GBs in transition

metal dichalcogenides for hydrogen evolution reaction³². However, the potential of GBs for modifying as well the selectivity of a reaction remains unexplored. In this work, we found that Au GBs exhibit high selectivity for the two-electron ORR due to their ~6% lattice expansion and coordination number of ~8, positioning their OOH binding energy near the top of the volcano plot. This results in nearly 100% selectivity for the two-electron ORR, showcasing the potential of GBs of modifying not only the catalytic activity, but also its selectivity towards a specific reaction.

4) *Stability of GBs in electrocatalysis:* There is no consensus in the literature on the stability of GBs in electrocatalysis, even for the same material and reaction. Some studies show that Pt GBs are stable during four-electron ORR,⁸ while others report that Pt GBs degrade easily due to high-energy states,⁹ with atoms at GBs dissolving first during four-electron ORR. This discrepancy might be due to variations in local composition at GBs. Hetero-atoms can reduce GB energy, enhancing thermodynamic stability.¹³ Trace impurities from synthesis processes, driven by thermodynamics, can segregate to GBs.¹³ These impurities, often light elements in low concentrations, are difficult to detect with conventional spectroscopy methods like XPS and EDS.¹⁴ Furthermore, they are often embedded within GBs, rendering many surface analysis techniques ineffective. Therefore, atomic-scale spatially resolved characterization techniques are needed to study local composition at GBs.

In our work, we utilized atom probe tomography (APT) to identify the local composition at nanoscale GBs. APT can detect and identify individual atoms within a sample, allowing precise determination of local composition at GBs.¹³ APT can quantify all elements in the periodic table, including light elements like hydrogen and boron, which are challenging to detect with other techniques. APT measurements show that boron species are segregated at the GBs of Au NAs. The two-electron ORR activity of Au NAs without B segregation degraded by 43.1% after 100 hours of durability testing, while Au NAs with 8% B segregation showed only a 3.8% degradation. The stability of the two-electron ORR activity is proportional to the B concentration at GBs. DFT calculations indicate that B segregation stabilizes Au GBs, as B atoms reduce the energy of the system, making GBs more stable and less prone to degradation. Therefore, our results highlight the importance of precise local composition characterization of nanomaterials, and proves that B segregation at GBs is a successful strategy for stabilizing these defects during electrocatalysis.

Due to the previously presented factors, we believe our work is novel and relevant enough for its publication in *Nature Communications*.

We have modified the Introduction to highlight better the importance of our work compared to other papers already published.

Page 1-2: “Grain boundaries (GBs) are two-dimensional structural defects that can effectively synergize these strategies: they can induce local lattice strain, atomic step ridges, and offer unique active sites for reactions like the four-electron oxygen reduction reaction (ORR). For instance, Maillard et al. demonstrated that Pt–Ni nanocrystals with increased surface distortion exhibit superior four-electron ORR activities compared to their less defective counterparts. However, due to the challenge of introducing GBs in nanocatalysts in a controlled manner, the understanding of how GBs modulate catalytic activity and their potential to catalyze other reactions remains limited. Existing strategies for tuning GB density modify also the size, shape and morphology of nanocatalysts, hampering the interpretation of the relationship between GBs and catalytic activity. Our understanding is also limited by incomplete characterization of GB type distribution, their atomic arrangement and local composition. This lack of understanding leads to a negligence of the potential of GBs to regulate not only the catalytic activity but also selectivity towards certain pathways. For instance, having an active catalyst for oxygen reduction that is also selective towards its two-electron pathway would enable the electrochemical production of hydrogen peroxide (H₂O₂), a chemical widely used in various industries.

In addition, the stability of GBs during electrocatalysis is also debated. Some studies suggest that Pt GBs are stable during four-electron ORR, while others report poor stability of GBs in polycrystalline PtCo nanowires compared to single-crystal counterparts. Atoms at GBs are thermodynamically unstable due to their high energy, which can lead to structural degradations and catalyst deactivation. Inconsistent stability reports may be due to local compositional variations and trace impurities at GBs, which are challenging to detect with conventional techniques but have been reported to effectively stabilize these defects. For instance, boron (B) is reported to stabilize the GBs of bulk polycrystals by lowering their energy, enhancing overall structural integrity. However, its effect on GB stabilization in nanocatalysts remains unexplored. Advanced atomic-scale characterization methods, such as atom probe tomography (APT), high-angle annular dark-field (HAADF) scanning transmission electron microscopy (STEM), and 4D-STEM, are essential for investigating the local composition and structure near GBs, thus advancing our understanding of their catalytic role.”

3. Detailed explanations of the Au nano-assembly synthesis mechanism are crucial. The authors need to address why Au nanoparticles coalesce spontaneously upon collision and illustrate the role of hydrogen gas bubbles in detail. If the bubbles only provide external forces, can other gases or methods (mechanical stirring, ultrasound) be used to achieve the same effect?

*Author reply: To illustrate the role of H₂ gas bubbles, we purged Ar and N₂ gases (instead of H₂) into the Au NP solution in a mildly alkaline environment (pH=10) for comparison. After 10 h of Ar gas purging, most Au NPs remained separated (**Figure R1a**), indicating that Ar gas purging is insufficient for assembling NPs into a network. FTIR analysis showed that all citrate ligands were removed from the Au NP surface (**Figure R1b**) due to destabilization in the mildly alkaline environment. Despite the absence of steric repulsion from citrate ligands, the NPs still failed to assemble into a network, suggesting the presence of additional inhibitory factors. ζ -potential measurements revealed that after 10 h of Ar gas purging, the ζ -potential of the Au NP solution remained at -33 mV (**Figure R1c**), indicating that the Au NP surfaces were still negatively charged, likely due to surface-bound OH⁻. This electrostatic repulsion from the negative charge of OH⁻ may explain why Ar gas purging cannot assemble NPs into a network.² Similarly, N₂ gas purging also failed to assemble NPs into a network (**Figure R1d-1f**).*

*In contrast, H₂ gas purging successfully resulted in the assembly of NPs into a network (**Figure R1g**). FTIR analysis confirmed the complete removal of citrate ligands from the NP surface (**Figure R1h**), and ζ -potential measurements showed that the ζ -potential was nearly zero (**Figure R1i**). This suggests that the NP surface charge transitioned from negative to nearly neutral, likely due to competitive adsorption between OH⁻ and hydrogen. As hydrogen atoms gradually occupied the binding sites where OH⁻ had been adsorbed, the NP surface became neutral, eliminating electrostatic repulsion. In comparison, Ar and N₂ gases, being inert, cannot adsorb onto the NP surface, leaving it negatively charged (**Figure R1c, 1f**).*

In summary, H₂ gas bubbling plays a dual role in the formation of Au NAs. First, purging an excess amount of hydrogen gas into the solution introduces competitive adsorption between OH⁻ and hydrogen. As hydrogen atoms gradually occupy the binding sites where OH⁻ was adsorbed, the negatively charged ligands are displaced and detached, removing electrostatic repulsion and transforming the negatively charged NP surface to an almost neutral state. Second, bubbling H₂ gas creates physical agitation in the solution, generating turbulence and convection currents. This physical disturbance increases the

frequency and energy of collisions between Au NPs. When two nanoparticles come into contact, they can coalesce and attach at regions where citrate molecules are absent, leading to the formation of GBs. Methods that only provide external forces, such as other gas bubbling, mechanical stirring, and ultrasound, are incapable of transforming the NP surface from a negative charge to a neutral state, thereby failing to eliminate electrostatic repulsion. Consequently, these methods cannot be employed to assemble NPs into a network structure.

Figure R1. (a) TEM image of the NPs obtained by purging Ar gas into a citrate-capped Au NP solution at pH 10 for 10 h. (b) FTIR spectrum of the Au NP solution at pH 10 subjected to Ar gas purging for 0 h and 10 h, respectively. (c) Time-dependent variation of the ζ -potential for citrate-capped Au NP solution at pH 10 during Ar gas purging. TEM image (d), FTIR spectrum (e) and time-dependent variation of ζ -potential (f) for citrate-capped Au NP solution at pH 10 during N₂ gas purging. TEM

image (g), FTIR spectrum (h) and time-dependent variation of ζ -potential (i) for citrate-capped Au NP solution at pH 10 during H₂ gas purging.

Regarding the question on why Au NPs coalesce spontaneously upon collision. As the ligands gradually detach from the surface of NPs, regions devoid of ligands become neutral and exposed. These bare regions lacking ligand capping result in the exposure of undercoordinated surface atoms, which exhibit high reactivity and energetically unfavorable characteristics due to their exposure to the surrounding environment. Seeking to minimize their energy, these undercoordinated surface atoms endeavor to form bonds with neighboring NPs. Upon the approach of two bare NPs, the elevated surface energy propels them to reduce their overall energy by diminishing the total surface area. Coalescence follows as the undercoordinated surface atoms on each NP bond together, thereby forming a GB.³³ This transformative process reduces the overall surface area and subsequently diminishes the total surface energy of the system, thus achieving thermodynamic stability. Moreover, the Van der Waals forces between the NPs can also contribute to their mutual attraction,² thereby facilitating coalescence, especially considering the neutral charge state of the NPs surface thanks to H substitution of OH⁻ ions. As the NPs draw closer, the significance of Van der Waals forces increases, fostering the formation of robust bonds between the surface atoms and ultimately promoting coalescence.²

Regarding the question on detailed explanations of the Au nano-assembly synthesis mechanism. The assembly of Au NPs comprises three distinct stages. Firstly, under alkaline conditions, the initially capping citrate ligands undergo gradual detachment from the NP surface due to their instability. Consequently, OH⁻ ions adsorb onto the NP surface. During this stage, the steric repulsion between Au NPs originating from the citrate ligands is alleviated, although electrostatic repulsion persists due to the adsorption of OH⁻ ions, as shown by ζ -potential measurements (**Figure R1c, R1f**).² Secondly, the continuous purging of an excess amount of hydrogen gas into the Au NP solution facilitates competitive adsorption between OH⁻ ions and H atoms. This results in the gradual replacement of surface-bound OH⁻ ions by H atoms, thereby transforming the negatively charged NP surface into a nearly neutral state and eliminating electrostatic repulsion between Au NPs (**Figure R1i**).² Thirdly, the introduction of H₂ gas into the NP solution induces turbulence and convection currents, thereby increasing the frequency of NP collisions. Upon approaching each other, two NPs can undergo spatial rotation, followed by attachment and coalescence at regions devoid of ligands to form a GB. The primary driving force for this spontaneous attachment is the elimination of bare regions lacking ligand protection,

possessing high surface energies, thereby reducing the total energy of the system.³ Concurrently, other structural defects such as step sites, stacking faults, and dislocations may arise due to imperfect orientation during the attachment process.³³ Furthermore, NP attachment typically results in anisotropic growth, giving rise to nanowire morphology.⁴

We added **Figure R1** as **Figure S6** to the supplementary information, and also added the following sentences to the main text.

Page 3: “The assembly of Au NPs comprises three distinct stages. Firstly, under mildly alkaline conditions (pH=10), the initially capped citrate ligands undergo gradual detachment from the NP surface (fig. S4) due to their instability (fig. S5), and as a consequence, OH⁻ ions can adsorb onto the NP surface. During this stage, the steric repulsion between Au NPs originating from the citrate ligands is alleviated, although electrostatic repulsion persists due to the adsorption of OH⁻ ions (fig. S6a-c). Secondly, the continuous bubbling of an excess amount of hydrogen gas into the Au NP solution facilitates competitive adsorption between OH⁻ ions and H atoms. This results in the gradual replacement of surface-bound OH⁻ ions by H atoms, thereby transforming the negatively charged NP surface into a nearly neutral state and eliminating electrostatic repulsion between Au NPs (fig. S6c). Thirdly, the introduction of H₂ gas into the NP solution induces turbulence and convection currents, thereby increasing the frequency of NP collisions. Upon approaching each other, two NPs can undergo spatial rotation, followed by attachment and coalescence at regions devoid of ligands to form a GB. The primary driving force for this spontaneous attachment is the elimination of bare regions lacking ligand protection, possessing high surface energies, thereby reducing the total energy of the system. Other structural defects such as step sites, stacking faults, and dislocations may arise due to imperfect orientation during the attachment process. Furthermore, NP attachment typically results in anisotropic growth, giving rise to nanowire morphology. If other gases (e.g., Ar, N₂) are used instead of H₂, the NP attachment does not occur (fig S6d-i) due to the electrostatic repulsions between OH⁻ capped NPs’ surfaces.”

4. The authors need to provide more characteristic results for the original Au NPs to demonstrate the grain boundary density before assembly.

Author reply: We would like to thank the reviewer for the suggestion. In the original version of the manuscript only one high resolution image of a Au NP is provided, not enough for assessing the GB density of the Au NPs before assembling.

In the revised version, we are including more high-resolution images of the original Au NPs. (Figure R5, Figure R6). As can be seen in Figure R5, most of the Au NPs are GB and defect free, and that is why they are assumed to have a GB surface density of 0. Additionally, the few NPs that do have some GBs, possess only low-energy twin $\Sigma 3$ GBs (Figure R6).

Figure R5. High resolution TEM images of Au NPs before assembling with corresponding indexed FFT. Notice how before the assembly process, most of the Au NPs are defect-free.

Figure R6. A trace number of Au NPs possess low energy $\Sigma 3$ GBs before assembling.

We added **Figure R5** and **Figure R6** as **Figure S12** and **Figure S11** in the supplementary information. Additionally, the following sentence has been added to the main text.

Page 5: “Although a trace number of Au NPs possess low-energy $\Sigma 3$ GBs (fig. S11), the vast majority of them are found to be free of GBs (fig. S12), implying that most of the GB defects are formed during the assembly process. Therefore, the GB density of Au NPs is set to $0 \mu\text{m}^{-1}$ in the follow-up discussion.”

5. In Fig. 1d, the Au NAs displayed many fivefold twin structures. According to Science 367.6473 (2020): 40-45, there should be two formation mechanisms for the fivefold twin structure. The authors should analyze the formation mechanism of Au NAs accordingly.

Author reply: We thank the reviewer for the suggestion. The formation of five-fold twin boundaries in Au NAs during the attachment of Au NPs can be attributed to two primary mechanisms: high-energy GB decomposition and partial dislocation slipping.³³

*High-energy GB decomposition is one mechanism that facilitates the formation of five-fold twin boundaries. When Au NPs coalesce, the initial interfaces often contain high-energy GBs, such as $\Sigma 9$, $\Sigma 11$, $\Sigma 27$, and $\Sigma 33$, due to misalignment of the crystal lattices (**Figure 1d**, **Figure S11**). To minimize the system's total energy, these high-energy GBs can decompose into a series of lower-energy twin boundaries.³⁴ In face-centered cubic (FCC) metals like gold, twin boundaries represent energetically favorable configurations.³⁵ This decomposition process can lead to the development of multiple twin planes that intersect at a common axis, forming a five-fold twin structure that stabilizes the system by reducing its overall energy.³³*

*Partial dislocation slipping is another pathway leading to the formation of five-fold twin boundaries.³⁶ During the attachment of Au NPs, partial dislocations can be introduced into the crystal lattice (**Figure 1d**). These partial dislocations traverse the lattice, creating stacking faults and twin planes as they slip.³³ When multiple partial dislocations move in a coordinated manner, they generate several twin planes that intersect at a common axis, resulting in a pentagonal symmetry where five twin planes converge to form a five-fold twin boundary. This configuration arises from the need to accommodate strain and reduce the system's total energy.*

Together, these mechanisms—high-energy GB decomposition and partial dislocation slipping—can explain the formation of five-fold twin boundaries in Au NAs during the attachment of Au NPs. These twin boundaries contribute to structural stability by minimizing surface and interfacial energies, resulting in a stable and energetically favorable five-fold symmetric structure.^{35, 37, 38}

To clarify this point, the following sentences have been added to the main text.

Page 3: “We also observed five-fold twin boundaries in Au NAs (Fig. 1D), likely formed through high-energy GB decomposition and partial dislocation slipping during NP attachment.³³ These twin boundaries enhance structural stability by minimizing surface and interfacial energies, resulting in a stable and energetically favorable five-fold symmetric structure.³³”

6. In Figs. 2f-h, the authors analyzed the strain of individual atoms by fitting 2D Gaussians to find the atomic column position. This result should be more reliable for a single phase (inside a single grain) since the atomic spacing at different phases and at the interface is incomparable. The method here should be further validated.

Author reply: We thank the reviewer for raising this point. As explained in the supplementary information, after finding the atomic column positions by fitting 2D Gaussians, we analyze the strain grain by grain, selecting the center of the grain as reference lattice (assumed to be unstrained). This is done for avoiding possible artefacts that could originate if different grains have a different zone axis, and thus different plane families are imaged (with different d-spacing).

We have rewritten the explanation in the supplementary information for making it clearer:

Page 3 in the supplementary information: “After having calculated the interatomic distances, the strain of individual grains was analyzed by selecting the center of the corresponding grain as

reference (unstrained) lattice. Afterwards, these strain maps of individual grains were combined for generating the complete strain map of the sample.”

Additionally, we have performed line scans in the HAADF-STEM micrographs for further validating the method. As can be seen in **Figure R7**, the same behavior is obtained as when performing Gaussian fitting.

Figure R7. (a) HAADF-STEM micrograph of Au NAs (same as in **Fig. 2** of main text). (b) Line profile along white arrow of (a). Notice how next to the $\Sigma 9$ GB, the interplanar distance is larger than at the center of the grain. This is in good agreement with the strain maps obtained by Gaussian fitting, in which tensile strain is measured along the yy direction in that region.

7. In Figs. 2j and k, there seems to be a stronger strain along the yy direction compared to the xx direction. This requires further explanation.

Author reply: The strain distribution of the sample is inhomogeneous, with some parts having more strain than others in a particular direction. The stronger strain along the yy direction is only observed in that particular region of the sample. In the strain maps of other regions provided in the supplementary information (**Figure S8-S10**), that behavior is not present. Therefore, we believe the stronger strain along the yy direction in **Figure 2** is just a consequence of the inhomogeneous distribution of the strain.

8. Regarding the lattice expansion induced by the grain boundary, the authors claim that "the lattice expansion derived from EXAFS analysis of L-Au NAs, M-Au NAs, and H-Au NAs are approximately 6%, 4%, and 2%, respectively." The discrepancy between lattice expansion and grain boundary density (H-Au NAs inducing the lowest expansion despite the highest grain boundary density) should be addressed.

*Author reply: We thank the reviewer for raising this point. We apologize for the error in the description of **Fig. 3j**. The lattice expansions derived from EXAFS analysis for L-Au NAs, M-Au NAs, and H-Au NAs were reported incorrectly (they were interchanged). The correct values are approximately 2%, 4%, and 6%, respectively. This error has been corrected in the revised manuscript.*

9. In Fig. 4b, why does the selectivity of Au NPs change with potential while Au NAs maintains the same selectivity? The authors should provide deeper discussion to demonstrate the superiority of the developed Au NAs catalysts.

*Author reply: We thank the reviewer for the suggestion. As the applied potential varies, there is a tendency for Au NPs to undergo aggregation (**Figure S23**), whereas Au NAs retain their original morphology without experiencing aggregation (**Figure S24**). This inclination for aggregation primarily stems from the under-coordinated atoms present on the surface of NPs, which demonstrate a higher propensity for aggregation than their more-coordinated counterparts.^{39, 40} Consequently, aggregation typically initiates from these under-coordinated surface sites of the Au NPs.*

*This aggregation process results in a reduced fraction of under-coordinated atoms exposed on the NP surface, as evidenced by a slight increase of 0.2 in the coordination number (CN) measured by EXAFS after the two-electron ORR test (**Table R2**). In contrast, the CN of Au NAs remains unchanged after the two-electron ORR test (**Table R2**). The network geometry of Au NAs, with multiple contact points with the carbon support, likely reduces motion and aggregation.⁴¹*

*DFT calculations show that CN significantly influences the two-electron ORR; when CN exceeds 8, the *OOH adsorption energy weakens with increasing CN (**Figure R4**). This weaker binding of the OOH intermediate reduces the activity and selectivity for the two-electron ORR pathway by increasing the activation barrier required for its reduction to H₂O₂.^{42, 43} Consequently, as aggregation progresses, fewer under-coordinated surface atoms remain exposed, leading to a decrease in selectivity for two-electron ORR. This explains why the selectivity of Au NPs changes with applied potential, while Au NAs maintain consistent selectivity.*

Table R2. EXAFS fitting results of Au L3 edge for Au NPs and H-Au NAs before and after ORR tests.

Sample	CN	R (Å)	σ^2 (10^{-3} Å ²)	ΔE_0 (eV)	R factor
Au NPs before ORR	11.8±0.2	2.62±0.02	2.9±1.4	4.4±1.2	0.013
Au NPs after ORR	12.0±0.1	2.62±0.03	2.5±1.2	4.1±1.7	0.011
H-Au NAs before ORR	8.7±0.2	2.78±0.02	2.8±1.2	3.5±1.5	0.012
H-Au NAs after ORR	8.7±0.2	2.78±0.03	3.1±1.3	3.7±1.3	0.013

CN: coordination number; R: bond length; σ : Debye-Waller factor; ΔE_0 : inner potential shift; R factor: goodness of the fit.

Figure R4. (a) Slab model of the Au (111) surface, highlighting the surface atom used for OOH binding energy calculations in deep blue. (b) Surface atom (deep blue) used for OOH binding energy calculations with different coordination numbers (CNs) on the Au (111) surface, showing only the

nearest neighbor atoms. (c) Calculated ORR activity volcano relationship between the limiting potential (U_L) and the free energy of *OOH ($\Delta G_{^*OOH}$) for the two-electron pathway to H_2O_2 . (d) Calculated reaction coordinate diagrams for Au with different CNs.

We added **Figure R4** as **Figure S21**, **Table R2** as **Table S3** in the supplementary information, and also added the following sentences to the main text.

Page 9: “As the applied potential varies, the selectivity for two-electron ORR in Au NPs diminishes, while Au NAs maintain consistent selectivity (Fig. 4B). This reduction in selectivity for Au NPs is due to the aggregation of under-coordinated surface atoms, which are more prone to aggregation than more-coordinated atoms, leading to fewer under-coordinated atoms with higher two-electron ORR selectivity remaining exposed (table S2-S3). In contrast, the coordination number of Au NAs remains unchanged after the two-electron ORR test (table S2-S3), likely because the network geometry of Au NAs and their multiple contact points with the carbon support reduce motion and aggregation tendencies.”

10. What are the grain boundary (GB) type distributions of L-Au NAs and M-Au NAs? Will the rate of hydrogen gas bubbles affect the GB type, and what is the impact of GB types on ORR performance?

*Author reply: We thank the reviewer for the suggestion. We have performed more 4D-STEM experiments to statistically characterize the GB type distributions in L-Au NAs and M-Au NAs. The analysis indicates that both L-Au NAs and M-Au NAs (**Figure R8**) have a similar GB type distribution to H-Au NAs (**Figure 1e-1g**), comprising approximately 10% low-angle GBs and 90% high-angle GBs (HAGBs), including about 30% $\Sigma 3$ GBs alongside other coincident site lattice theory GBs. Since the rate of hydrogen gas bubbles did not affect the GB type distribution, the changes in activity can be attributed to the other structural changes (i.e., coordination number, GB surface density and lattice expansion).*

*To investigate the impact of GB type on two-electron ORR performance, we constructed three representative models of Au GBs commonly found in all nanoassembly samples: $\Sigma 3$ GB, $\Sigma 9$ GB, and $\Sigma 27$ GB, to simulate catalytic sites (**Figure R9a**). The ORR activity volcano plot was calculated using the limiting potential (U_L) as the performance indicator (**Figure R9b**), defined as the maximum potential at which both the one-electron reduction of O_2 to *OOH and the subsequent one-electron reduction of *OOH to H_2O_2 are energetically favorable. On a GB-free Au surface, the *OOH adsorption energy is located on the right side of the volcano plot (**Figure R9b**), indicating a low affinity for *OOH adsorption*

that impedes O_2 adsorption and slows the reaction, reflected in a small U_L . In contrast, GB atoms significantly increase $*OOH$ adsorption strength, placing it on the left side of the volcano plot (**Figure R9b**), which reflects a stronger affinity for $*OOH$ compared to GB-free sites. The $\Sigma 3$ GB model shows a minimal free energy deviation of -0.01 eV from the ideal state, positioning the $*OOH$ adsorption energy at the peak of the volcano plot. In comparison, the $*OOH$ adsorption energies for the $\Sigma 9$ GB and $\Sigma 27$ GB models exceed the optimal value (**Figure R9b-9c**), which could adversely affect two-electron ORR selectivity.

Figure R8. (a) HAADF-STEM image, (b) in-plane and (c) out-of-plane orientation maps from the corresponding 4D-STEM dataset for L-Au NAs. (d) HAADF-STEM image and grain orientation maps

(*e*: in-plane, *f*: out-of-plane) from the corresponding 4D-STEM dataset for M-Au NAs. (*g*) Histogram plots of the GB types derived from over 100 GBs in the 4D-STEM data.

Figure R9. (a) Top and side views of the models used to calculate the OOH binding energy on Au $\Sigma 3$ GB, $\Sigma 9$ GB, and $\Sigma 27$ GB. The surface atom used in these calculations is highlighted in blue. (b)

Calculated ORR activity volcano plot showing the relationship between the limiting potential (U_L) and the free energy of *OOH ($\Delta G_{^*OOH}$) for the two-electron pathway to H_2O_2 on Au $\Sigma 3$ GB, $\Sigma 9$ GB, and $\Sigma 27$ GB. (c) Calculated reaction coordinate diagrams for Au $\Sigma 3$ GB, $\Sigma 9$ GB, and $\Sigma 27$ GB.

We added **Figure R8** as **Figure S19** and **Figure R9** as **Figure S22** to the supplementary information, and the following sentences to the main text.

Page 6: “4D-STEM measurements indicate that both L-Au NAs and M-Au NAs have a similar GB type distribution to H-Au NAs, comprising approximately 10% LAGBs and 90% HAGBs, including about 30% $\Sigma 3$ GBs alongside other CSL GBs. The rate of hydrogen gas bubbles did not affect the GB type distribution.”

Page 10: “GBs also play a crucial role in two-electron ORR activity (fig. S22). On GB-free Au surfaces, the *OOH adsorption energy is too low (positioned on the right side of the volcano plot, fig. S22b), signifying low *OOH affinity and restricted O_2 adsorption. In contrast, GB atoms have a significantly higher *OOH adsorption energy, which falls on the left side of the volcano plot, reflecting a higher affinity compared to GB-free sites (fig. S22b). The $\Sigma 3$ GB model, in particular, shows a minimal free energy deviation of -0.01 eV from the ideal state (fig. S22c), placing *OOH adsorption energy at the volcano plot's peak. Conversely, *OOH adsorption energies for $\Sigma 9$ GB and $\Sigma 27$ GB models exceed the optimal range, which may facilitate too much O_2 dissociation and adversely affect two-electron ORR selectivity.”

Page 12: “The experimental assessments of the d-band center, coupled with theoretical computations, consistently validate that lattice expansion, optimized coordination number and GBs synergistically contribute to the enhancement of the two-electron ORR performance by fine-tuning the adsorption strength of the reaction intermediate *OOH on the surface of Au.”

11. For the stability concerning the B species, the authors claimed that "After the durability testing, the concentration of B at the GBs of H-Au NAs remains nearly constant, whereas in L-Au NAs, B is nearly gone. This observation further underlines that the concentration of B species at GBs is a primary contributor to the superior stability exhibited by H-Au NAs." However, the change in B content seems more likely to be a result rather than a cause. In other words, it should be the instability of the sample that leads to the loss of B, rather than the loss of B causing a decrease in stability. Therefore, there is

still no direct evidence to suggest a direct association between B species and sample stability. Furthermore, the impact of B species on stability seems to deviate from the main theme of the article, which focuses on grain boundary engineering.

*Author reply: We thank the reviewer for raising this point. To investigate whether B segregation stabilizes GBs, B was intentionally excluded during synthesis by using citric acid rather than boric acid to produce B-free Au NAs using a high flow rate of H₂ gas (B-free H-Au NAs). B-free Au NAs exhibit a highly porous network where NP building blocks are interconnected predominantly by $\Sigma 3$ GBs (**Figure R10**), similar in morphology (**Figure R10a**, **Figure 1c**), lattice expansion degree (**Table R3**), and coordination number (**Table R3**) to H-Au NAs. The initial two-electron ORR activity of B-free H-Au NAs (**Figure R11a-11b**) is comparable to that of H-Au NAs (**Figure 4a-4b**), indicating that B segregation is not responsible for the high two-electron ORR activity of H-Au NAs. After 100 hours of two-electron ORR durability testing, the current of B-free H-Au NAs decayed by 43.1% (**Figure R11c-11d**), while H-Au NAs showed only a 3.8% decay (**Figure 5a-5b**). Post-durability characterization showed a marked increase in the coordination number and a significant decrease in lattice distance of B-free H-Au NAs (**Table R3**), indicating substantial structural changes. These significant differences in stability suggests that B plays a crucial role in preserving the catalyst structure, and therefore high catalytic activity during two-electron ORR catalysis.*

*DFT calculations indicate that the increased coordination number (**Figure R4**) and lattice contraction (**Figure 4e, 4f**) lead to a higher activation barrier for the two-electron ORR, reducing activity and selectivity. In contrast, the coordination number and lattice distance of H-Au NAs remained unchanged after the durability test (**Table R3**), further underscoring the importance of B in maintaining structural integrity. Compared to B-free H-Au NAs, Au NAs with B segregation at GBs demonstrated improved durability in two-electron ORR. Even trace amounts of B at the GBs resulted in notable stability improvements. For example, L-Au NAs with minimal B segregation showed a 22.1% current decay after 100 hours (**Figure 5a-5b**), significantly better than the stability of B-free H-Au NAs (**Figure R11c-11d**). The greater the segregation of B at GBs, the better the stability of Au NAs in two-electron ORR (**Figure 5a-5b**, **Figure R11c-11d**).*

*To further understand the effect of B on GB stability, we performed DFT simulations using a B-segregated Au $\Sigma 3$ GB model as a function of B concentration (**Figure R12a**). By varying B concentrations, we computed the GB strengthening energy ($E_{\text{strengthening}}$), where a negative $E_{\text{strengthening}}$ indicates enhanced GB strength and a positive $E_{\text{strengthening}}$ signifies weakened GB strength. The results*

showed that $E_{\text{strengthening}}$ becomes increasingly negative as the B concentration at the GBs increases (Figure R12b), suggesting stronger GBs that are less prone to migration and annihilation. Additionally, B atoms may act as pinning agents, hindering GB movement and thus stabilizing the GB structure.⁴⁴ Therefore, it is reasonable to infer that B segregation stabilizes the GBs in Au NAs, but a sufficient amount of B is required for long-term stability.

We added Figure R10, Figure R11, Figure R12 as Figure S33, Figure S34, Figure S36, Table R3 as Table S5 to the supplementary information, and also added the following sentences to the main text.

Page 12: “To determine whether B segregation contributes to the high two-electron ORR activity and durability, B was intentionally excluded during the synthesis to produce B-free Au NAs under a high flow rate of H₂ gas of 300 sccm (denoted as B-free H-Au NAs). B-free Au NAs exhibit a highly porous network where NP building blocks are interconnected predominantly by $\Sigma 3$ GBs (fig. S33), similar in morphology (fig. S33a, Fig. 1c), lattice expansion (table S5), and coordination number (table S5) to H-Au NAs. The two-electron ORR activity of B-free Au NAs (fig. S34a-b) is comparable to that of H-Au NAs (Fig5A,B), indicating that B segregation does not account for the high two-electron ORR activity observed in H-Au NAs (fig. S35). Durability tests for two-electron ORR revealed a current decay of 43.1% for B-free H-Au NAs after 100 hours (fig. S34c-d), whereas H-Au NAs showed a significantly lower decay of only 3.8% (Fig 5A,B). Post-durability tests characterization indicates substantial structural changes in B-free H-Au NAs, marked by an increased coordination number and decreased lattice expansion (table S5). In contrast, H-Au NAs structure was maintained, highlighting the essential role of B in enhancing structural stability and consequent ORR durability. To further explore the impact of B on GB stability, we conducted DFT simulations using a B-segregated Au $\Sigma 3$ GB model at varying B concentrations (fig. S36a). The results demonstrated that increasing B concentration at GBs leads to more negative GB strengthening energy ($E_{\text{strengthening}}$; fig. S36b), suggesting that higher B concentrations enhance GB strength, reducing their propensity for migration and annihilation. This finding underscores the necessity of adequate B concentration for maintaining structural stability and sustained ORR durability.”

Figure R10. (a) Low-magnification HAADF-STEM image of B-free H-Au NAs. (b) HAADF-STEM image and (c, d) grain orientation maps from the corresponding 4D-STEM dataset for B-free H-Au NAs (c: in-plane orientation, d: out-of-plane orientation). (e) Histogram plots of the GB types derived from analysis of over 100 grains of the 4D-STEM data.

Table R3. EXAFS fitting results of Au L3 edge for B-free H-Au NAs and H-Au NAs before and after ORR tests.

Sample	CN	R (\AA)	σ^2 (10^{-3}\AA^2)	ΔE_0 (eV)	R factor
B-free H-Au NAs before ORR	8.8 ± 0.3	2.76 ± 0.02	3.4 ± 1.6	4.2 ± 1.5	0.019
B-free H-Au NAs after ORR	9.6 ± 0.2	2.70 ± 0.03	3.2 ± 1.4	3.9 ± 1.2	0.015
H-Au NAs before ORR	8.7 ± 0.2	2.78 ± 0.02	2.8 ± 1.2	3.5 ± 1.5	0.012
H-Au NAs after ORR	8.7 ± 0.2	2.78 ± 0.03	3.1 ± 1.3	3.7 ± 1.3	0.013

CN: coordination number; R: bond length; σ : Debye-Waller factor; ΔE_0 : inner potential shift; R factor: goodness of the fit.

Figure R11. (a) Linear sweep voltammetry of B-free H-Au NAs recorded at 1600 rpm and a scan rate of 5 mV s^{-1} in 0.1 M HClO_4 , together with the detected H_2O_2 currents on the ring electrode at a fixed potential of 1.2 V vs. RHE . (b) Comparison of the calculated H_2O_2 selectivity of B-free H-Au NAs and H-Au NAs during a potential sweep. (c) Durability measurements of B-free H-Au NAs at a fixed disk potential of 0.35 V . (d) Comparison of the calculated H_2O_2 selectivity of B-free H-Au NAs and H-Au NAs after durability tests.

Figure R12. (a) Top and side views of the models for the calculation of the strengthening energy ($E_{\text{strengthening}}$) of the B-segregated Au $\Sigma 3$ GB using the following equation: $E_{\text{strengthening}} = (E_{\text{GB+B}} - E_{\text{GB}}) - (E_{\text{FS+B}} - E_{\text{FS}})$, where E_{GB} and $E_{\text{GB+B}}$ represent the total energies of the B-segregated Au $\Sigma 3$ twin GB. Similarly, E_{FS} and $E_{\text{FS+B}}$ denote the total energies of the Au free surface without and with B segregation, respectively. (b) $E_{\text{strengthening}}$ of the B-segregated Au $\Sigma 3$ GB at varying B concentrations. A negative $E_{\text{strengthening}}$ indicates that the GB is strengthened, whereas a positive $E_{\text{strengthening}}$ signifies that the GB is weakened.

Reviewer 3

Comments:

The paper entitled “Grain Boundary Engineering for Efficient and Durable Electrocatalysis” submitted to Nature Communication focuses on the beneficial role on grain boundaries (GB) in the surface adsorption properties of Au nanoparticles. The authors successfully tuned the density of GB while maintaining a similar size of the coherent domains of Au nanoparticles. This work is of high interest to the electrocatalysis community. I recommend publication after addressing the following comments in a revised version of the manuscript. By doing so, the conclusions will certainly be strengthened.

Author reply: We extend our appreciation to the reviewer for dedicating the time to evaluate our work and providing valuable insights.

1. My Main concern relates to the role of boron in activity/stability relationships. First, boron is only mentioned at the end of the abstract “Their structural and catalytic stability can be attributed to segregated boron at the GBs...” without any word of its origin. Only at pages 9/15, it is learnt that boron comes from boric acid used in the nanoparticle’s synthesis process. Furthermore, quantitatively speaking, "traces amounts of boron" are mentioned, which I find misleading for the reader. In Figure 5 and S25, the 3D atom mapping shows a boron content of around 5-10 at% located in the vicinity of the GBs. Such content is substantial! The authors should clarify the role of boron because it likely also influences the reactivity of the Au nanoparticles by modifying the electronic structure of Au.

Author reply: We would like to thank the reviewer for raising these three points.

To provide further clarity, more discussion of the role of B in stabilizing GBs has been included in the Introduction:

Page 2: “For instance, boron (B) is reported to stabilize the GBs of bulk polycrystals by lowering their energy, enhancing overall structural integrity. However, its effect on GB stabilization in nanocatalysts remains unexplored.”

Additionally, the origin of B is also explained in the Introduction

Page 2: “... (stemming from boric acid used during the synthesis)...”

Regarding the question of B concentration, B is primarily located at the GB regions, where its concentration is 8 at.%. However, the overall boron concentration in the H-Au NAs is only 0.57 at.%, as measured by APT. XPS provides only the overall composition without spatial resolution at the GBs,

resulting in a low detected B concentration. Precise quantification of low concentrations of B using XPS is challenging due to its low atomic number ($Z=5$), which leads to the emission of low-energy characteristic X-rays that are difficult to detect and easily absorbed by the sample or detector window.

To clarify this point, we have added the following sentences to the main text:

Page 10-11: “Furthermore, trace amounts of boron have been found within Au NAs by XPS (fig. S26). However, the precision of XPS in quantifying B and its spatial resolution are limited, posing challenges to achieving precise quantitative analyses.”

Page 11: “Despite the high B concentration in the GBs, the overall B concentrations in L-Au NAs, M-Au NAs, and H-Au NAs are smaller, of 0.14 at.%, 0.36 at.%, and 0.57 at.%, respectively.”

In response to the question on the effect of B on reactivity, to investigate whether B segregation is the cause of high two-electron ORR activity, B was intentionally excluded during the synthesis by using citric acid rather than boric acid to produce B-free Au NAs under a high flow rate of H₂ gas of 300 sccm (denoted as B-free H-Au NAs). B-free Au NAs exhibit a highly porous network where NP building blocks are interconnected predominantly by $\Sigma 3$ GBs (**Figure R10**), similar in morphology (**Figure R10a**, **Figure 1c**), lattice expansion (**Table R3**), and coordination number (**Table R3**) to H-Au NAs. The two-electron ORR activity of B-free H-Au NAs (**Figure R13**) was comparable to that of H-Au NAs (**Figure 4a-4b**), indicating that B segregation is not responsible for the high two-electron ORR activity of H-Au NAs.

To further understand the effect of B on GB activity, DFT simulations were performed using Au $\Sigma 3$ GB, $\Sigma 9$ GB, and $\Sigma 27$ GB slab models with and without segregated boron atom (**Figure R14a**). The volcano plot was calculated using the limiting potential (U_L) as the performance indicator, defined as the maximum potential at which both the one-electron reduction of O₂ to *OOH and the subsequent one-electron reduction of *OOH to H₂O₂ are energetically favorable. The simulation results show that the *OOH binding energy remains almost unchanged with or without segregated B atom (**Figure R14b-14c**), indicating that B segregation does not significantly affect two-electron ORR activity. Thus, the influence of B on the two-electron ORR activity of Au NAs is negligible.

We added **Figure R10**, **Figure R13**, **Figure R14** as **Figure S33**, **Figure S34a-b**, **Figure S35**, **Table R4** as **Table S5** to the supplementary information, and also added the following sentences to the main text.

Page 11-12: “To determine whether B segregation contributes to the high two-electron ORR activity and durability, B was intentionally excluded during the synthesis to produce B-free Au NAs under a high flow rate of H₂ gas of 300 sccm (denoted as B-free H-Au NAs). B-free Au NAs exhibit a highly porous network where NP building blocks are interconnected predominantly by $\Sigma 3$ GBs (fig. S33), similar in morphology (fig. S33a, Fig. 1c), lattice expansion (table S5), and coordination number (table S5) to H-Au NAs. The two-electron ORR activity of B-free Au NAs (fig S34a,b) is comparable to that of H-Au NAs (Fig. 5A,B), indicating that B segregation does not account for the high two-electron ORR activity observed in H-Au NAs (fig. S35).”

Figure R10. (a) Low-magnification HAADF-STEM image of B-free H-Au NAs. (b) HAADF-STEM image and (c, d) grain orientation maps from the corresponding 4D-STEM dataset for B-free H-Au NAs (c: in-plane orientation, d: out-of-plane orientation). (e) Histogram plots of the GB types derived of the analysis from over 100 GBs grains of the 4D-STEM data.

Table R4. EXAFS fitting results of Au L3 edge for B-free H-Au NAs and H-Au NAs before ORR tests.

Sample	CN	R (Å)	σ^2 (10^{-3}Å^2)	ΔE_0 (eV)	R factor
B-free H-Au NAs before ORR	8.8 ± 0.3	2.76 ± 0.02	3.4 ± 1.6	4.2 ± 1.5	0.019
H-Au NAs before ORR	8.7 ± 0.2	2.78 ± 0.02	2.8 ± 1.2	3.5 ± 1.5	0.012

CN: coordination number; R: bond length; σ : Debye-Waller factor; ΔE_0 : inner potential shift; R factor: goodness of the fit.

Figure R13. (a) Linear sweep voltammetry of B-free H-Au NAs recorded at 1600 rpm and a scan rate of 5 mV s^{-1} in 0.1 M HClO_4 , together with the detected H_2O_2 currents on the ring electrode at a fixed

potential of 1.2 V vs. RHE. (b) The calculated H_2O_2 selectivity of B-free H-Au NAs during a potential sweep.

Figure R14. (a) Top and side views of the models used to calculate the OOH binding energy on B-segregated Au $\Sigma 3$ GB, $\Sigma 9$ GB, and $\Sigma 27$ GB, with the surface Au atom for OOH binding energy calculations highlighted in deep blue and the segregated B atom shown in green. (b) Calculated ORR activity volcano plot illustrating the relationship between the limiting potential (U_L) and the free energy

of *OOH (ΔG_{*OOH}) for the two-electron pathway to H_2O_2 on Au $\Sigma 3$ GB, $\Sigma 9$ GB, and $\Sigma 27$ GB, both with and without B segregation. (c) Calculated reaction coordinate diagrams for Au $\Sigma 3$ GB, $\Sigma 9$ GB, and $\Sigma 27$ GB, comparing scenarios with and without B segregation.

2. Following up on my initial comment, it would have been interesting to disentangle the roles of boron and structural defects. In the case of “classical” Au nanoparticles, boron is located at the surface, whereas for the NAs Au with no boron detected on the surface but at the vicinity of the GBs, the comparison of activity and stability trends is more complicated. Atom probe tomography results indicate a clear relationship between boron content and oxygen reduction activity. Since boron content and GB density are linked, it is unclear if the authors can claim that GBs (and structural defects) enhance activity while the presence of boron improves the stability of GBs. Can the authors synthesize Au nanoparticles without using boric acid or by replacing it with another type of acid?

Author reply: We are grateful to the reviewer for offering this insightful suggestion. After 100 hours of two-electron ORR durability testing, the current of B-free H-Au NAs decayed by 43.1% (**Figure R14a-14b**), while H-Au NAs showed only a 3.8% decay (**Figure 5a-5b**). This significant difference in stability suggests that B plays a crucial role in enhancing the durability of two-electron ORR. Post-durability characterization showed a marked increase in the coordination number and a significant decrease in lattice distance of B-free H-Au NAs (**Table R3**), suggesting substantial structural changes. DFT calculations indicate that the increased coordination number (**Figure R4**) and lattice contraction (**Figure 4e, 4f**) lead to a higher activation barrier for the two-electron ORR, reducing activity and selectivity. In contrast, the structure of H-Au NAs remained unchanged after the durability test (**Table R3**), further underscoring the importance of B in maintaining structural integrity. Compared to B-free H-Au NAs, Au NAs with B segregation at GBs demonstrated improved durability in two-electron ORR. Even trace amounts of B at the GBs resulted in notable stability improvements. For example, L-Au NAs with minimal B segregation showed a 22.1% current decay after 100 hours (**Figure 5a-5b**), significantly better than the stability of B-free H-Au NAs (**Figure R14a-14b**). The greater the segregation of B at GBs, the better the stability of Au NAs in two-electron ORR (**Figure 5a-5b, Figure R14a-14b**).

To further understand the effect of B on GB stability, we performed DFT simulations using a B-segregated Au $\Sigma 3$ GB model as a function of B concentration (**Figure R12a**). By varying B concentrations, we computed the GB strengthening energy ($E_{\text{strengthening}}$), where a negative $E_{\text{strengthening}}$ indicates enhanced GB strength and a positive $E_{\text{strengthening}}$ signifies weakened GB strength. The results

showed that $E_{\text{strengthening}}$ becomes increasingly negative as the B concentration at the GBs increases (Figure R12b), suggesting stronger GBs that are less prone to migration and annihilation. Additionally, B atoms may act as pinning agents, hindering GB movement and thus stabilizing the GB structure. Therefore, it is reasonable to infer that B segregation stabilizes the GBs in Au NAs, but a sufficient amount of B is required for long-term stability.

We added Figure R12, Figure R15 as Figure S36, Figure S34c-d, Table R3 as Table S5 to the supplementary information, and also added the following sentences to the main text.

Page 12: “Durability tests for two-electron ORR revealed a current decay of 43.1% for B-free H-Au NAs after 100 hours (fig. S34 c,d), whereas H-Au NAs showed a significantly lower decay of only 3.8% (Fig 5A,B). Post-durability tests characterization indicates substantial structural changes in B-free H-Au NAs, marked by an increased coordination number and decreased lattice expansion (table S5). In contrast, H-Au NAs structure was maintained, highlighting the essential role of B in enhancing structural stability and consequent ORR durability. To further explore the impact of B on GB stability, we conducted DFT simulations using a B-segregated Au Σ 3 GB model at varying B concentrations (fig. S36a). The results demonstrated that increasing B concentration at GBs leads to more negative GB strengthening energy ($E_{\text{strengthening}}$, fig. S36b), suggesting that higher B concentrations enhance GB strength, reducing their propensity for migration and annihilation. This finding underscores the necessity of adequate B concentration for maintaining structural stability and sustained ORR durability.”

Figure R15. (a) Durability measurements of B-free H-Au NAs at a fixed disk potential of 0.35 V. (b) The calculated H₂O₂ selectivity of B-free H-Au NAs after durability tests.

a Au $\Sigma 3$ (111) twin GB model with B segregation

Au (111) free surface model with B segregation

Au (111) free surface model without B segregation

Side View

Top View

Figure R12. (a) Top and side views of the models for the calculation of the strengthening energy ($E_{\text{strengthening}}$) of the B-segregated Au $\Sigma 3$ GB using the following equation: $E_{\text{strengthening}} = (E_{\text{GB+B}} - E_{\text{GB}}) - (E_{\text{FS+B}} - E_{\text{FS}})$, where E_{GB} and $E_{\text{GB+B}}$ represent the total energies of the B-segregated Au $\Sigma 3$ twin GB.

Similarly, E_{FS} and E_{FS+B} denote the total energies of the Au free surface without and with B segregation, respectively. (b) $E_{strengthening}$ of the B-segregated Au $\Sigma 3$ GB at varying B concentrations. A negative $E_{strengthening}$ indicates that the GB is strengthened, whereas a positive $E_{strengthening}$ signifies that the GB is weakened.

Table R3. EXAFS fitting results of Au L3 edge for B-free H-Au NAs and H-Au NAs before and after ORR tests.

Sample	CN	R (Å)	σ^2 (10^{-3} Å ²)	ΔE_0 (eV)	R factor
B-free H-Au NAs before ORR	8.8±0.3	2.76±0.02	3.4±1.6	4.2±1.5	0.019
B-free H-Au NAs after ORR	9.6±0.2	2.70±0.03	3.2±1.4	3.9±1.2	0.015
H-Au NAs before ORR	8.7±0.2	2.78±0.02	2.8±1.2	3.5±1.5	0.012
H-Au NAs after ORR	8.7±0.2	2.78±0.03	3.1±1.3	3.7±1.3	0.013

CN: coordination number; R: bond length; σ : Debye-Waller factor; ΔE_0 : inner potential shift; R factor: goodness of the fit.

3. The role of structural defects in electrocatalysis of the oxygen reduction reaction has been studied by L. Dubau and F. Maillard. I recommend that the following publications be mentioned in this manuscript: Nat. Mater. 17 (2018) 827-833, ACS Catal., 7 (2017) 3072-3081, and ACS Catal., 6 (2016) 4673-4684.

Author reply: We appreciate the reviewer's suggestion and have added the mentioned publications to the manuscript.

To clarify this point, the following sentences are added to the main text, together with the aforementioned references.

Page 1-2: "Grain boundaries (GBs) are two-dimensional structural defects that can effectively synergize these strategies: they can induce local lattice strain,^{23, 45, 46} atomic step ridges,⁴⁷ and offer unique active sites for reactions like the four-electron oxygen reduction reaction (ORR).^{8, 9, 48-50} For instance, Maillard et al. demonstrated that Pt–Ni nanocrystals with increased surface

distortion exhibit superior four-electron ORR activities compared to their less defective counterparts.⁴⁸⁻⁵⁰

References

1. Moglianetti, M.; Solla-Gullón, J.; Donati, P.; Pedone, D.; Debellis, D.; Sibillano, T.; Brescia, R.; Giannini, C.; Montiel, V.; Feliu, J. M., Citrate-coated, size-tunable octahedral platinum nanocrystals: a novel route for advanced electrocatalysts. *ACS applied materials & interfaces* **2018**, *10* (48), 41608-41617.
2. Matter, F.; Luna, A. L.; Niederberger, M., From colloidal dispersions to aerogels: How to master nanoparticle gelation. *Nano Today* **2020**, *30*, 100827.
3. Zhang, H.; Banfield, J. F., Energy calculations predict nanoparticle attachment orientations and asymmetric crystal formation. *The Journal of Physical Chemistry Letters* **2012**, *3* (19), 2882-2886.
4. Zhu, C.; Peng, H.-C.; Zeng, J.; Liu, J.; Gu, Z.; Xia, Y., Facile synthesis of gold wavy nanowires and investigation of their growth mechanism. *Journal of the American Chemical Society* **2012**, *134* (50), 20234-20237.
5. Nguyen, K. T.; Hiep Vuong, V.; Nguyen, T. N.; Nguyen, T. T.; Yamamoto, T.; Hoang, N. N., Unusual hydrogen implanted gold with lattice contraction at increased hydrogen content. *Nature communications* **2021**, *12* (1), 1560.
6. Hinnemann, B.; Moses, P. G.; Bonde, J.; Jørgensen, K. P.; Nielsen, J. H.; Horch, S.; Chorkendorff, I.; Nørskov, J. K., Biomimetic hydrogen evolution: MoS₂ nanoparticles as catalyst for hydrogen evolution. *Journal of the American Chemical Society* **2005**, *127* (15), 5308-5309.
7. Antonov, V.; Antonova, T.; Belash, I.; Gorodezkii, A.; Ponyatovskii, E. In *Synthesis of the gold hydride under hydrogen high pressure*, Dokl. Acad. Nauk SSSR, 1982; pp 376-380.
8. Zhu, E.; Xue, W.; Wang, S.; Yan, X.; Zhou, J.; Liu, Y.; Cai, J.; Liu, E.; Jia, Q.; Duan, X., Enhancement of oxygen reduction reaction activity by grain boundaries in platinum nanostructures. *Nano Research* **2020**, *13*, 3310-3314.
9. Kabiraz, M. K.; Ruqia, B.; Kim, J.; Kim, H.; Kim, H. J.; Hong, Y.; Kim, M. J.; Kim, Y. K.; Kim, C.; Lee, W.-J., Understanding the grain boundary behavior of bimetallic platinum-cobalt alloy nanowires toward oxygen electro-reduction. *ACS Catalysis* **2022**, *12* (6), 3516-3523.
10. Zhang, Z.; Luo, Z.; Chen, B.; Wei, C.; Zhao, J.; Chen, J.; Zhang, X.; Lai, Z.; Fan, Z.; Tan, C., One-Pot Synthesis of Highly Anisotropic Five-Fold-Twinned PtCu Nanoframes Used as a Bifunctional Electrocatalyst for Oxygen Reduction and Methanol Oxidation. *Advanced Materials (Deerfield Beach, Fla.)* **2016**, *28* (39), 8712-8717.
11. Huang, H.; Ruditskiy, A.; Choi, S.-I.; Zhang, L.; Liu, J.; Ye, Z.; Xia, Y., One-pot synthesis of penta-twinned palladium nanowires and their enhanced electrocatalytic properties. *ACS applied materials & interfaces* **2017**, *9* (36), 31203-31212.
12. Zhang, T.; Bai, Y.; Sun, Y.; Hang, L.; Li, X.; Liu, D.; Lyu, X.; Li, C.; Cai, W.; Li, Y., Laser-irradiation induced synthesis of spongy AuAgPt alloy nanospheres with high-index facets, rich grain boundaries and subtle lattice distortion for enhanced electrocatalytic activity. *Journal of Materials Chemistry A* **2018**, *6* (28), 13735-13742.
13. Kim, S.-H.; Yoo, S.-H.; Chakraborty, P.; Jeong, J.; Lim, J.; El-Zoka, A. A.; Zhou, X.; Stephenson, L. T.; Hickel, T.; Neugebauer, J. r., Understanding alkali contamination in colloidal nanomaterials to unlock grain boundary impurity engineering. *Journal of the American Chemical Society* **2022**, *144* (2), 987-994.
14. Kim, S. H.; Yoo, S. H.; Shin, S.; El-Zoka, A. A.; Kasian, O.; Lim, J.; Jeong, J.; Scheu, C.; Neugebauer, J.; Lee, H., Controlled doping of electrocatalysts through engineering impurities. *Advanced Materials* **2022**, *34* (28), 2203030.
15. Xia, C.; Xia, Y.; Zhu, P.; Fan, L.; Wang, H., Direct electrosynthesis of pure aqueous H₂O₂ solutions up to 20% by weight using a solid electrolyte. *Science* **2019**, *366* (6462), 226-231.

16. Seh, Z. W.; Kibsgaard, J.; Dickens, C. F.; Chorkendorff, I.; Nørskov, J. K.; Jaramillo, T. F., Combining theory and experiment in electrocatalysis: Insights into materials design. *Science* **2017**, *355* (6321), eaad4998.
17. Yang, S.; Verdager-Casadevall, A.; Arnarson, L.; Silvioli, L.; Colic, V.; Frydendal, R.; Rossmeis, J.; Chorkendorff, I.; Stephens, I. E., Toward the decentralized electrochemical production of H₂O₂: a focus on the catalysis. *Acs Catalysis* **2018**, *8* (5), 4064-4081.
18. Da Rosa, G.; Maugis, P.; Portavoce, A.; Drillet, J.; Valle, N.; Lentzen, E.; Hoummada, K., Grain-boundary segregation of boron in high-strength steel studied by nano-SIMS and atom probe tomography. *Acta Materialia* **2020**, *182*, 226-234.
19. Tytko, D.; Choi, P.-P.; Klöwer, J.; Kostka, A.; Inden, G.; Raabe, D., Microstructural evolution of a Ni-based superalloy (617B) at 700 C studied by electron microscopy and atom probe tomography. *Acta materialia* **2012**, *60* (4), 1731-1740.
20. Feng, B.; Yokoi, T.; Kumamoto, A.; Yoshiya, M.; Ikuhara, Y.; Shibata, N., Atomically ordered solute segregation behaviour in an oxide grain boundary. *Nature communications* **2016**, *7* (1), 11079.
21. Yang, C.; Wang, Y.; Sigle, W.; van Aken, P. A., Determination of grain-boundary structure and electrostatic characteristics in a SrTiO₃ bicrystal by four-dimensional electron microscopy. *Nano Letters* **2021**, *21* (21), 9138-9145.
22. Liu, X.-J.; Yin, X.; Sun, Y.-D.; Yu, F.-J.; Gao, X.-W.; Fu, L.-J.; Wu, Y.-P.; Chen, Y.-H., Interlaced Pd–Ag nanowires rich in grain boundary defects for boosting oxygen reduction electrocatalysis. *Nanoscale* **2020**, *12* (9), 5368-5373.
23. Huang, W.; Johnston-Peck, A. C.; Wolter, T.; Yang, W.-C. D.; Xu, L.; Oh, J.; Reeves, B. A.; Zhou, C.; Holtz, M. E.; Herzing, A. A., Steam-created grain boundaries for methane C–H activation in palladium catalysts. *Science* **2021**, *373* (6562), 1518-1523.
24. Feng, X.; Jiang, K.; Fan, S.; Kanan, M. W., Grain-boundary-dependent CO₂ electroreduction activity. *Journal of the American Chemical Society* **2015**, *137* (14), 4606-4609.
25. Wang, J.-Q.; Xi, C.; Wang, M.; Shang, L.; Mao, J.; Dong, C.-K.; Liu, H.; Kulinich, S. A.; Du, X.-W., Laser-generated grain boundaries in ruthenium nanoparticles for boosting oxygen evolution reaction. *ACS Catalysis* **2020**, *10* (21), 12575-12581.
26. Nagaoka, Y.; Suda, M.; Yoon, I.; Chen, N.; Yang, H.; Liu, Y.; Anzures, B. A.; Parman, S. W.; Wang, Z.; Grünwald, M., Bulk grain-boundary materials from nanocrystals. *Chem* **2021**, *7* (2), 509-525.
27. Zhao, X.; Gunji, T.; Lv, F.; Huang, B.; Ding, R.; Liu, J.; Luo, M.; Zou, Z.; Guo, S., Direct observation of heterogeneous surface reactivity and reconstruction on terminations of grain boundaries of platinum. *ACS Materials Letters* **2021**, *3* (5), 622-629.
28. She, X.; Zhu, X.; Yang, J.; Song, Y.; She, Y.; Liu, D.; Wu, J.; Yu, Q.; Li, H.; Liu, Z., Grain-boundary surface terminations incorporating oxygen vacancies for selectively boosting CO₂ photoreduction activity. *Nano Energy* **2021**, *84*, 105869.
29. Li, Y.; Cui, F.; Ross, M. B.; Kim, D.; Sun, Y.; Yang, P., Structure-sensitive CO₂ electroreduction to hydrocarbons on ultrathin 5-fold twinned copper nanowires. *Nano letters* **2017**, *17* (2), 1312-1317.
30. Li, T.; Kasian, O.; Cherevko, S.; Zhang, S.; Geiger, S.; Scheu, C.; Felfer, P.; Raabe, D.; Gault, B.; Mayrhofer, K. J. J., Atomic-scale insights into surface species of electrocatalysts in three dimensions. *Nature Catalysis* **2018**, *1* (4), 300-305.
31. Chen, Z.; Wang, T.; Liu, B.; Cheng, D.; Hu, C.; Zhang, G.; Zhu, W.; Wang, H.; Zhao, Z.-J.; Gong, J., Grain-boundary-rich copper for efficient solar-driven electrochemical CO₂ reduction to ethylene and ethanol. *Journal of the American Chemical Society* **2020**, *142* (15), 6878-6883.
32. He, Y.; Tang, P.; Hu, Z.; He, Q.; Zhu, C.; Wang, L.; Zeng, Q.; Golani, P.; Gao, G.; Fu, W., Engineering grain boundaries at the 2D limit for the hydrogen evolution reaction. *Nature communications* **2020**, *11* (1), 57.
33. Song, M.; Zhou, G.; Lu, N.; Lee, J.; Nakouzi, E.; Wang, H.; Li, D., Oriented attachment induces fivefold twins by forming and decomposing high-energy grain boundaries. *Science* **2020**, *367* (6473), 40-45.

34. Wolf, D., Structure-energy correlation for grain boundaries in FCC metals—III. Symmetrical tilt boundaries. *Acta metallurgica et materialia* **1990**, *38* (5), 781-790.
35. Barnard, A. S.; Young, N. P.; Kirkland, A. I.; Van Huis, M. A.; Xu, H., Nanogold: a quantitative phase map. *ACS nano* **2009**, *3* (6), 1431-1436.
36. Zhu, Y.; Liao, X.; Valiev, R., Formation mechanism of fivefold deformation twins in nanocrystalline face-centered-cubic metals. *Applied physics letters* **2005**, *86* (10).
37. Young, N.; Van Huis, M.; Zandbergen, H.; Xu, H.; Kirkland, A., Transformations of gold nanoparticles investigated using variable temperature high-resolution transmission electron microscopy. *Ultramicroscopy* **2010**, *110* (5), 506-516.
38. Koga, K.; Ikeshoji, T.; Sugawara, K.-i., Size- and temperature-dependent structural transitions in gold nanoparticles. *Physical review letters* **2004**, *92* (11), 115507.
39. Tang, L.; Han, B.; Persson, K.; Friesen, C.; He, T.; Sieradzki, K.; Ceder, G., Electrochemical stability of nanometer-scale Pt particles in acidic environments. *Journal of the American Chemical Society* **2010**, *132* (2), 596-600.
40. Tang, L.; Li, X.; Cammarata, R. C.; Friesen, C.; Sieradzki, K., Electrochemical stability of elemental metal nanoparticles. *Journal of the American Chemical Society* **2010**, *132* (33), 11722-11726.
41. Li, M.; Zhao, Z.; Cheng, T.; Fortunelli, A.; Chen, C.-Y.; Yu, R.; Zhang, Q.; Gu, L.; Merinov, B. V.; Lin, Z., Ultrafine jagged platinum nanowires enable ultrahigh mass activity for the oxygen reduction reaction. *Science* **2016**, *354* (6318), 1414-1419.
42. Ren, X.; Dong, X.; Liu, L.; Hao, J.; Zhu, H.; Liu, A.; Wu, G., Research progress of electrocatalysts for the preparation of H₂O₂ by electrocatalytic oxygen reduction reaction. *SusMat* **2023**, *3* (4), 442-470.
43. Wang, K.; Huang, J.; Chen, H.; Wang, Y.; Song, S., Recent advances in electrochemical 2e oxygen reduction reaction for on-site hydrogen peroxide production and beyond. *Chemical Communications* **2020**, *56* (81), 12109-12121.
44. Buban, J.; Matsunaga, K.; Chen, J.; Shibata, N.; Ching, W.; Yamamoto, T.; Ikuhara, Y., Grain boundary strengthening in alumina by rare earth impurities. *Science* **2006**, *311* (5758), 212-215.
45. Gsell, M.; Jakob, P.; Menzel, D., Effect of substrate strain on adsorption. *Science* **1998**, *280* (5364), 717-720.
46. Mariano, R. G.; McKelvey, K.; White, H. S.; Kanan, M. W., Selective increase in CO₂ electroreduction activity at grain-boundary surface terminations. *Science* **2017**, *358* (6367), 1187-1192.
47. Mariano, R. G.; Kang, M.; Wahab, O. J.; McPherson, I. J.; Rabinowitz, J. A.; Unwin, P. R.; Kanan, M. W., Microstructural origin of locally enhanced CO₂ electroreduction activity on gold. *Nature Materials* **2021**, *20* (7), 1000-1006.
48. Chattot, R.; Le Bacq, O.; Beermann, V.; Kühl, S.; Herranz, J.; Henning, S.; Kühn, L.; Asset, T.; Guétaz, L.; Renou, G., Surface distortion as a unifying concept and descriptor in oxygen reduction reaction electrocatalysis. *Nature materials* **2018**, *17* (9), 827-833.
49. Dubau, L.; Nelayah, J.; Asset, T.; Chattot, R. I.; Maillard, F. d. r., Implementing structural disorder as a promising direction for improving the stability of PtNi/C nanoparticles. *ACS Catalysis* **2017**, *7* (4), 3072-3081.
50. Dubau, L.; Nelayah, J.; Moldovan, S.; Ersen, O.; Bordet, P.; Drnec, J.; Asset, T.; Chattot, R.; Maillard, F., Defects do catalysis: CO monolayer oxidation and oxygen reduction reaction on hollow PtNi/C nanoparticles. *ACS Catalysis* **2016**, *6* (7), 4673-4684.

REVIEWERS' COMMENTS

Reviewer #1 (Remarks to the Author):

The manuscript has been revised by fully reflecting the reviewers' comments. The current version looks publishable as is.

Reviewer #2 (Remarks to the Author):

I have reviewed the revised version of this work. The authors have addressed the questions in the previous review and provided sufficient additional experimental and simulation results to support their claims. Therefore, I recommend to accept this work.

Reviewer #3 (Remarks to the Author):

The authors have carefully taken into account all my comments. The paper can now be accepted.

Response to Reviewers' Comments

Ms. ID: NCOMMS-24-15705-A

Title: Grain Boundary Engineering for Efficient and Durable Electrocatalysis

Reviewers' comments in normal type

Author response in italics

Reviewer 1

Comments:

The manuscript has been revised by fully reflecting the reviewers' comments. The current version looks publishable as is.

Author reply: We thank the reviewer for the positive opinion about our work.

Reviewer 2

Comments:

I have reviewed the revised version of this work. The authors have addressed the questions in the previous review and provided sufficient additional experimental and simulation results to support their claims. Therefore, I recommend to accept this work.

Author reply: We express our gratitude to the reviewer for the positive assessment of our research.

Reviewer 3

Comments:

The authors have carefully taken into account all my comments.

The paper can now be accepted.

Author reply: We appreciate the reviewer's positive assessment of our work.